

Retrieving the global distribution of threshold of wind erosion from satellite data and

implementing it into the GFDL AM4.0/LM4.0 model

Bing Pu[1,2,*], Paul Ginoux[2], Huan Guo[2,3], N. Christine Hsu[4], John Kimball[5], Beatrice

Marticorena[6], Sergey Malyshev[2], Vaishali Naik[2], Norman T. O'Neill[7], Carlos Pérez

García-Pando[8], Joseph M. Prospero[9], Elena Shevliakova[2], Ming Zhao[2]

[1]Atmospheric and Oceanic Sciences Program, Princeton University,

Princeton, New Jersey 08544

[2] NOAA Geophysical Fluid Dynamics Laboratory, Princeton, New Jersey 08540

[3] Cooperative Programs for the Advancement of Earth System Science, University

Corporation for Atmospheric Research, Boulder, Colorado, 80301

[4] NASA Goddard Space Flight Center, Greenbelt, Maryland, 20771

[5] Department of Ecosystem and Conservation Sciences, University of Montana,

Missoula, Montana 59812

[6]LISA, Universités Paris Est-Paris Diderot-Paris

[7] Département de géomatique appliquée , Université de Sherbrooke

[8] Barcelona Supercomputing Center, Barcelona, Spain, 08034

[9] Rosenstiel School of Marine and Atmospheric Sciences, University of Miami, Miami,

Florida, 33149

[*] Current affiliation: Department of Geographical and Atmospheric Science, the

University of Kansas, Lawrence, Kansas, 66045





**Abstract.** Dust emission is initiated when surface wind velocities exceed the threshold
of wind erosion. Most dust models used constant threshold values globally. Here we use
satellite products to characterize the frequency of dust events and surface properties. By
matching this frequency derived from Moderate Resolution Imaging Spectroradiometer
(MODIS) Deep Blue aerosol products with surface winds, we are able to retrieve a
climatological monthly global distribution of wind erosion threshold ($V_{threshold}$) over dry
and sparsely-vegetated surface. This monthly two-dimensional threshold velocity is then
implemented into the Geophysical Fluid Dynamics Laboratory coupled land-atmosphere
model (AM4.0/LM4.0). It is found that the climatology of dust optical depth (DOD) and
total aerosol optical depth, surface $PM_{10}$ dust concentrations, and seasonal cycle of DOD
are better captured over the "dust belt" (i.e. North Africa and the Middle East) by
simulations with the new wind erosion threshold than those using the default globally
constant threshold. The most significant improvement is the frequency distribution of
dust events, which is generally ignored in model evaluation. By using monthly rather
than annual mean $V_{threshold}$, all comparisons with observations are further improved. The
monthly global threshold of wind erosion can be retrieved under different spatial
resolutions to match the resolution of dust models and thus can help improve the
simulations of dust climatology and seasonal cycle as well as dust forecasting.



## 1. Introduction

Mineral dust is one of the most abundant aerosols by mass and plays an important
role in the climate system. Dust particles absorb and scatter solar and terrestrial radiation,
thus modifying local energy budget and consequently atmospheric circulation patterns.
Studies have shown that the radiative effect of dust can affect a wide range of
environmental processes. Dust is shown to modulate West African (e.g., Miller and
Tegen, 1998; Miller et al., 2004; Mahowald et al., 2010; Strong et al., 2015) and Indian
(e.g., Jin et al., 2014; Vinoj et al., 2014; Jin et al., 2015; Jin et al., 2016; Solmon et al.,
2015; Kim et al., 2016; Sharma and Miller, 2017) monsoonal precipitation. During severe
droughts in North America, there is a positive feedback between dust and the
hydrological cycle (Cook et al., 2008, 2009; 2013). African dust is also found to affect
Atlantic tropical cyclone activities (e.g., Dunion and Velden, 2004; Wong and Dessler,
2005; Evan et al., 2006; Strong et al., 2018). When deposited on snow and ice, dust
reduces the surface reflectivity, enhancing net radiant energy loading and accelerating
snow and ice melting, and consequently affecting runoff (e.g., Painter et al., 2010; 2018;
Dumont et al., 2014). Dust can serve as ice nuclei and affect the formation, lifetime, and
characteristic of clouds (e.g., Levin et al., 1996; Rosenfield et al., 1997; Wurzler et al.,
2000; Nakajima et al., 2001; Bangert et al., 2012), perturbing the hydrological cycle. Iron
and phosphorus enriched dust is also an important nutrient for the marine and terrestrial
ecosystems and thus interacts with the ocean and land biogeochemical cycles (e.g., Fung
et al., 2000; Jickells et al., 2005; Shao et al., 2011; Bristow et al., 2010; Yu et al., 2015).
Given the importance of mineral dust, many climate models incorporate dust
emission schemes to simulate the life cycle of dust (e.g., Donner et al., 2011; Collins et



al., 2011; Watanabe et al., 2011; Bentsen et al., 2013).  Mineral dust particles are lifted
from dry and bare soils into the atmosphere by saltation and sandblasting. This process is
initiated when surface winds reach a threshold velocity of wind erosion. The value of this
wind erosion threshold depends on soil and surface characteristics, including soil
moisture, soil texture and particle size, and presence of pebbles, rocks, and vegetation
residue (e.g., Gillette et al., 1980; Gillette and Passi, 1988; Raupach et al., 1993; Fécan et
al., 1999; Zender et al., 2003; Mahowald et al., 2005), and thus varies spatially and
temporally (Helgren and Prospero, 1987). Due to a lack of in-situ data at global scale and
uncertainties on these dependencies, most dust and climate models prescribe a spatially
and temporally constant threshold of wind erosion for simplicity.  Globally uniform
values (e.g., around 6 to 6.5 m s$^{-1}$) are either directly used over dry surfaces (e.g., Tegen
and Fung, 1994; Takemura et al., 2000; Uno et al., 2001; Donner et al., 2011) or with
modulations related to other factors, such as soil moisture (e.g.,  Takemura et al., 2000;
Ginoux et al. 2001; Zender et al., 2003; Kok et al., 2014a). For instance, in the
Geophysical Fluid Dynamics Laboratory coupled land-atmosphere model AM4.0/LM4.0
(Zhao et al., 2018a, b), a constant threshold of 6 m s$^{-1}$ is used.

The threshold of wind erosion may be approximately inferred using observations.

For instance, Chomette et al. (1999) used the Infrared Difference Dust Index (IDDI) and
10 m winds reanalysis from the European Centre for Medium-Range Weather Forecasts
(ECMWF) between 1990 and 1992 to calculate the threshold of wind erosion over seven
sites over the Sahel and Sahara. The IDDI was used to determine whether there was a
dust event for subsequently calculating an emission index defined as the number of dust
events to the total number of potential events. The distribution of surface wind speed was



matched with the emission index, and the threshold of wind erosion was determined
when the emission index was around 0.9. The resulting average threshold of wind erosion
ranged from 6.63 m s$^{-1}$ at a Sahelian site to about 9.08 m s$^{-1}$ at a Niger site, consistent
with the model results by Marticorena et al. (1997).

Later, Kurosaki and Mikami (2007) used World Meteorological Organization

(WMO) station data from March 1998 to June 2005 to examine the threshold wind speed
in East Asia. Using the distribution of surface wind speed and associated weather
conditions (i.e., with or without dust emission events), they approximated a dust emission
frequency by dividing number of dust events to the total number of observations for each
wind bin, and then determined threshold wind speeds at the 5% and 50% levels,
corresponding to the most favorable and normal land surface conditions for dust
emission, respectively.  They found that the derived threshold wind speed varied in space
and time, with a larger seasonal cycle in grassland regions, such as northern Mongolia,
and smaller seasonal variations in desert regions, such as the Taklimakan and Gobi
Deserts and the Loess Plateau.  Cowie et al. (2014) applied a similar method over
northern Africa, using wind data observed between 1984 and 2012, and focused on
threshold winds at the 25%, 50%, and 75% levels.

Draxler et al. (2010) derived the distribution of threshold of wind erosion over the

U.S. by matching the frequency of occurrence (FoO) of Moderate Resolution Imaging
Spectroradiometer (MODIS) Deep Blue (Hsu et al., 2004) aerosol optical depth (AOD)
above 0.75 with the FoO of friction velocities extracted from the North American
Mesoscale (NAM) forecast model at each grid point. This new threshold and a soil
characteristics factor was then incorporated into the Hybrid Single-Particle Lagrangian



Integrated Trajectory (HYSPLIT) model  (Draxier and Hess, 1998) to forecast dust
surface concentrations. It was found that major observed dust plume events in June and
July 2007 were successfully captured by the model. Later, Ginoux and Deroubaix (2017)
used FoO derived from  the MODIS Deep Blue dust optical depth (DOD) record to
retrieve the wind erosion threshold over East Asia.

For individual dust events, the threshold of friction velocity can also be

determined by fitting a second-order Taylor series to dust saltation flux measurements
(Barchyn and Hugenholtz, 2011; Kok et al., 2014b).

Nonetheless, a global distribution of threshold of wind erosion based on

observation that may be implemented in climate models is still lacking. In this study, we
propose a method to retrieve monthly global threshold of wind erosion (hereafter,
$V_{threshold}$) for dry and sparsely-vegetated surface using high-resolution satellite products
and reanalysis datasets.  This two-dimensional threshold is then implemented into the
Geophysical Fluid Dynamics Laboratory (GFDL) coupled land-atmosphere model,
AM4.0/LM4.0 (Zhao et al., 2018a, b). The benefits of using this threshold in simulating
present-day climatology and seasonal cycles of dust are analyzed by comparing the
model results with observations.

The data and method used to retrieve the threshold of wind erosion are detailed in

section 2. The distribution of the derived $V_{threshold}$ and its implication in the climate model
is presented in section 3. Section 4 discusses the uncertainties associated with this
method, and major conclusions are summarized in section 5.




## 2. Data and Methodology

### 2.1 Data

#### 2.1.1 Satellite products

1) MODIS Aqua and Terra dust optical depth

DOD is column-integrated extinction by mineral particles. Here daily DOD is retrieved from MODIS Deep Blue aerosol products (collection 6, level 2; Hsu et al., 2013; Sayer et al., 2013): aerosol optical depth (AOD), single-scattering albedo, and the Ångström exponent. All the daily variables are first interpolated to a 0.1° by 0.1° grid using the algorithm described by Ginoux et al. (2010). We require that the single-scattering albedo at 470 nm to be less than 1 for dust due to its absorption of solar radiation. This separates dust from scattering aerosols, such as sea salt. Then a continuous function relating the Ångström exponent, which is highly sensitive to particle size (Eck et al., 1999), to fine-mode AOD established by Anderson et al. (2005; their Eq. 5) is used to separate dust from fine particles. Details about the retrieval process and estimated errors are summarized by Pu and Ginoux (2018b). High-resolution MODIS DOD products have been used to identify and characterize dust sources (Ginoux et al., 2012; Baddock et al., 2016) and examine the variations of dustiness in different regions (e.g., Pu and Ginoux, 2016, 2017, 2018b).

Following the recommendation from Baddock et al. (2016), who found the dust sources are better detected using DOD with a low-quality flag (i.e., QA=1) than that with a high-quality flag (i.e., QA=3) as retrieved aerosol products were poorly flagged over dust source regions, we also use DOD with the flag of QA=1. Both daily DOD retrieved from Aqua and Terra platforms are used by averaging the two when both



products are available or using either one when only one product is available.  Since
Terra passes the equator from north to south around 10:30 am local time (LT) and Aqua
passes the Equator from south to north around 13:30 pm LT, an average of the two
combines the information from both morning and afternoon hours. This process also
largely reduces missing data (Pu and Ginoux, 2018b). This combined daily DOD,
hereafter MODIS DOD, is available from January 2003 to December 2015 at a resolution
of 0.1° by 0.1° grid.

2) Soil moisture

Soil moisture is an important factor that affects dust emission (Fécan et al., 1999).

Daily surface volumetric soil moisture (VSM) retrievals derived from similar calibrated
microwave (10.7 GHz) brightness temperature observations from the Advanced
Microwave Scanning Radiometer-Earth Observing System (AMSR-E) onboard the
NASA Aqua satellite (from June 2002 to October 2011) and the Advanced Microwave
Scanning Radiometer 2 (AMSR2) sensor onboard the JAXA GCOM-W1 satellite  (from
July 2012 to June 2017) from the University of Montana (Du et al., 2017a; Du et al.,
2017b) was used. Both AMSR-E and AMSR2 sensors provide global measurements of
polarized microwave emissions at six channels, with ascending and descending orbits
crossing the equator at around 1:30 pm and 1:30 am LT, respectively. The VSM
retrievals are derived from an iterative retrieval algorithm that exploits the variable
sensitivity of different microwave frequencies and polarizations, and minimizes the
potential influence of atmosphere, vegetation, and surface water cover on the soil signal.
The VSM record represents surface (top ~2 cm) soil conditions and shows favorable



global accuracy and consistent performance (Du et al. 2017b), particularly over areas
with low to moderate vegetation cover that are also more susceptible to wind erosion.
The horizontal resolution of the product is about 25 km by 25 km, and the daily product
from January 2003 to December 2015 is used. The ascending and descending obit VSM
retrievals are averaged to get the mean VSM for each day.

3) Snow cover

Snow cover may affect dust emission in the mid-latitudes during spring, for

instance, over northern China (Ginoux and Deroubaix, 2017). The interannual variation
of snow cover is also found to affect dust emission in regions, such as Mongolia
(Kurosaki and Mikami, 2004).  Here monthly snow cover data from MODIS/Terra level
3 data (Hall and Riggs, 2015) with a resolution of 0.05° by 0.05° from 2003 to 2015 is
used. The high spatial resolution of the product is very suitable for this study.

4) Leaf area index (LAI)

Vegetation can protect soil from the effects of wind and thus modulate dust

emission (e.g., Marticorena and Bergametti, 1995; Zender et al., 2003). While dense
vegetation coverage can increase surface roughness and reduce near surface wind speed,
the roots of vegetation can increase soil cohesion and further reduce wind erosion.  LAI
describes the coverage of vegetation with a unit of $m^2/m^2$, i.e., leaf area per ground area.
Here monthly LAI retrieved by Boston University from MODIS onboard Aqua (via
personal communication with Ranga Myneni and Taejin Park; Boston University, 2016)
with a resolution of 0.1° by 0.1° from 2003 to 2015 is used.



**2.1.2 Reanalysis**

Surface wind speed is a critical factor that affects wind erosion. Here 6 hourly 10

m wind speed from the NCEP/NCAR reanalysis (Kalnay et al., 1996, hereafter NCEP1)
on a T62 Gaussian grid (i.e., 192 longitude grids equally spaced and 94 latitude grids
unequally spaced) is used.   The NCEP1 is a global reanalysis with relatively long
temporal coverage, from 1948 to the present. We chose to use the NCEP1 reanalysis also
because surface winds in the GFDL AM4.0 model are nudged toward the NCEP1, and we
preferred to use the reanalysis surface wind that is closet to the model climatology.

ERA-Interim (Dee et al., 2011) is another global reanalysis produced from

ECMWF. It provides high spatial resolution (about 0.75° or 80 km) 6-hourly, daily, and
monthly reanalysis from 1979 to present day. Here we use soil temperature from the
ERA-Interim to determine the regions where wind erosion may be prohibited by the
frozen surface. Monthly temperature of the first soil layer (0 to 0.07 m) from 2003 to
2015 is used.

**2.1.3 Station data**
1) AERONET

The AErosol RObotic NETwork (AERONET; Holben et al., 1998) provides

quality assured cloud-screened (level 2) aerosol measurements from sunphotometer
records. In this paper we used the data products of the version 3.0 AERONET processing
routine. To examine model simulated DOD, we used coarse mode AOD (COD) at 500
nm processed by the Spectral Deconvolution Algorithm (O'Neill et al., 2003; hereafter
SDA). SDA COD monthly data is first screened to remove those months with less than



five days of records. To get the annual means, years with less than five months of records
were removed. Only stations with records of at least three years during the period were
used to calculate the 2003-2015 climatology (the same time period when MODIS DOD is
available). Overall, records from 313 stations were obtained.

AERONET monthly aerosol optical thickness (AOT) data around 550 nm (e.g.,

500 nm, 551 nm, 531 nm, 440 nm, 675 nm, 490 nm, 870 nm, etc.) and the Ångström
exponents across the dual wavelength of 440-675 nm, 440-870 nm, and 500-870 nm are
used to calculate AOD at 550 nm ($\tau_{550}$). If AOT for 551 nm, 555 nm, 531 nm or 532 nm
exist, then these values are directly used as AOD 550 nm. Otherwise, the AOT at
wavelength $\lambda_A$ (less than 550 nm), i.e., $\tau_A$ , AOT at wavelength $\lambda_B$ (larger than 550 nm),
i.e., $\tau_B$, and Ångström exponent between wavelengths $\lambda_A$ and $\lambda_B$ ($\alpha$) are used to derive
AOD 550 nm using the following equations:
$$\tau_{550} = \tau_A \left(\frac{550}{\lambda_A}\right)^{-\alpha} \qquad \text{if } \tau_A \text{ is available} ,\qquad (1)$$

$$\tau_{550} = \tau_B \left(\frac{550}{\lambda_B}\right)^{-\alpha} \qquad \text{if } \tau_B \text{ is available.}\qquad (2)$$

While this process of extrapolating to 550 nm using a classical Ångström exponent is a
bit incoherent with the higher order spectral approach of the SDA, errors due to the
choice of spectral order will be negligible in comparison with the types of model versus
measurement differences that we will be evaluating in this paper.

In a manner similar to the process of screening SDA COD data, monthly AOD

550 nm data with less than three days of records in a given month are removed. When
calculating the annual means we excluded years having less than five months of records.



Finally, to calculate the climatology of 2003-2015, only stations with at least three years
of records during this period are used totaling to 351.
We also developed a method to derive DOD at 550 nm from AOD 550 nm based
on the relationship between Ångström exponent and fine-mode AOD established by
Anderson et al. (2005; their Eq. 5). This adds a few more sites over the Sahel than the
SDA COD stations. DOD is calculated by subtracting the fine-mode AOD from the total
AOD. Due to the large uncertainties of single scattering albedo in AERONET records
over regions where AOD is lower than 0.4 (e.g., Dubovik and King, 2000; Holben et al.,
2006; Andrews et al., 2017), we did not use single scattering albedo to screen AOD to
further separate dust from scattering aerosols. Therefore, the derived AERONET DOD
over coastal stations may be contaminated by sea salt.

2) RSMAS surface dust concentration
The Rosenstiel School of Marine and Atmospheric Science (hereafter RSMAS
dataset) at University of Miami collected mass concentration of dust, sea salt, and sulfate
over stations globally, with most of stations on islands (Savoie and Prospero, 1989). The
dataset has been widely used for model evaluation (e.g., Ginoux et al., 2001; Huneeus et
al., 2011).
Only stations with records longer than four years were used and of those stations
only those years with at least eight months of data are used for calculating climatological
annual means. So, totally 16 stations are used. Station names and locations are listed in
Table S1 of the Supplement. We compare the climatology of annual mean surface dust
concentration with model output during 2000-2015. Most station records end earlier than





1998. So here we also assume that the climatology of the surface dust concentrations do
not change greatly from the 1980s to the 2000s.

3) IMPROVE surface fine dust concentration

The Interagency Monitoring of Protected Visual Environments (IMPROVE)

network has collected near-surface particulate matter 2.5 ($PM_{2.5}$) samples in the U.S.
since 1988 (Malm et al., 1994; Hand et al., 2011). IMPROVE stations are located in
national parks and wilderness areas, and $PM_{2.5}$ sampling is performed twice weekly
(Wednesday and Saturday; Malm et al., 1994) prior to 2000 and every third day
afterwards. Fine dust (with aerodynamic diameter less than 2.5 μm) concentration is
calculated using the concentrations of aluminum (Al), silicon (Si), calcium (Ca), iron
(Fe), and titanium (Ti) by assuming oxide norms associated with predominant soil
species (Malm et al., 1994; their Eq. 5). This dataset has been widely used to study
variations in surface fine dust in the U.S. (e.g., Hand et al., 2016; Hand et al., 2017, Tong
et al., 2017; Pu and Ginoux, 2018a). Here only monthly data with at least 50% of daily
data available in a month (i.e., at least 5 records) are used. Since station coverage over the
central U.S. increases after 2002 (e.g., Pu and Ginoux, 2018a), monthly station data from
2002 to 2015 are used and interpolated to a 0.5° by 0.5° grid using inverse distance
weighting interpolation. The gridded data are used to evaluate modeled surface fine dust
concentrations.

4) LISA $PM_{10}$ surface concentration





Surface $PM_{10}$ concentration from stations from the Sahelian Dust Transect, which

was deployed in 2006 under the framework of African Monsoon Multidisciplinary
Analysis International Program (Marticorena et al., 2010), were used to examine the
surface dust concentration over the Sahelian region. The data are maintained by
Laboratoire Interuniversitaire des Systèmes Atmosphériques (LISA) in the framework of
the International Network to study Deposition and Atmospheric composition in Africa
(INDAAF; Service National d'Observation de l'Institut National des Sciences de
l'Univers, France) network. Three stations are located within the pathway of Saharan and
Sahelian dust plumes moving towards the Atlantic Ocean. Here hourly $PM_{10}$
concentrations from these stations, Banizoumbou (Niger, 13.54° N, 2.66° E), Cinzana
(Mali, 13.28° N, 5.93° W), and M'Bour (Senegal, 14.39° N, 16.96° W), from 2006 to
2014 are used. The hourly station data are averaged to obtain daily and monthly mean
records to compare with model output.

**2.1.4 Other data**

Soil depth from the Food and Agriculture Organization of the United Nations

(FAO/IIASA/ISRIC/ISS-CAS/JRC, 2009) on a 0.08° by 0.08° resolution is used to
examine whether the soil depth is too shallow (i.e. less than 15 cm) for wind erosion.

**2.2 Retrieving threshold of wind erosion**

The monthly climatological threshold of wind erosion is retrieved by matching

the frequency distribution of the MODIS DOD at certain level with the frequency



distribution of surface 10 m winds from the NCEP1 reanalysis over the period from 2003
to 2015.  The process can be summarized by the following steps:
Step1: Since dust is emitted from the dry and sparsely-vegetated surface, the daily DOD
data is first masked out to remove the influences of non-erodible factors and unfavorable
environmental conditions that are known to prevent dust emission using criteria as
follows: daily VSM less than 0.1 $cm^3$ $cm^{-3}$; monthly LAI less than 0.3; monthly snow
cover less than 0.2%; monthly top-layer soil temperature higher than 273.15 K, i.e., over
unfrozen surface; and soil depth thicker than 15 cm.

Similar criteria have been used in previous studies to detect or confine dust source

regions. For instance, Kim et al. (2013) used NDVI less than 0.15, soil depth greater than
10 cm, surface temperature greater than 260 K, and without snow cover to mask
topography based dust source function. LAI less than 0.3 has been used as a threshold for
dust emission in the Community Land Model (Mahowald et al., 2010; Kok et al., 2014a),
while gravimetric soil moisture ranging from 1.01 to 11.2 kg $kg^{-3}$ depending on soil clay
content is recommended to constrain dust emission (Fécan et al., 1999).
Step 2: Masked daily DOD from Step 1 is then interpolated to a 0.5° by 0.5° grid using
bilinear interpolation. This is close to the horizontal resolution of the GFDL
AM4.0/LM4.0 model used in this study. Then the cumulative frequency distribution of
daily DOD from 2003 to 2015 is derived at each grid point for each month.
Step 3: Daily maximum surface wind speed is first derived from 6-hourly NCEP1 surface
winds and then interpolated to a 0.5° by 0.5° grid. The cumulative frequency distribution
of daily maximum surface wind from 2003 to 2015 is then calculated at each grid point
for each month.





Step 4: A minimum value of DOD ($DOD_{thresh}$) is used to separate dust events from
background dust. The cumulative frequency (in %) of dust events passing this threshold
is compared to the cumulative frequency of surface winds. The minimum surface winds
with the same frequency correspond to the threshold of wind erosion, $V_{threshold}$ (see a
schematic diagram in Figure S1 in the Supplement). This operation is performed for all
grid points for each month. Ginoux et al. (2012) used $DOD_{thresh} = 0.2$ to quantify the FoO
of local dust events. Similarly, $DOD_{thresh} = 0.2$ is used here in major dusty regions (North
Africa, Middle East, India, northern China), while for less dusty regions, such as the U.S.,
South America, South Africa, and Australia, $DOD_{thresh} = 0.02$ is used. The reason to use a
lower $DOD_{thresh}$ for less dusty regions is because: i) the overall dust emission in these
regions are at least ten times smaller than major dusty regions, such as North Africa (e.g.,
Huneeus et al., 2011); ii) the frequency distribution of DOD in these regions also peaks at
a much lower DOD band (see discussion in section 3.3).

Figures 1a-e show the seasonal and annual mean FoO (days when DOD is greater

than $DOD_{thresh}$) using the $DOD_{thresh}$ defined here. The shaded area covers major dust
sources, and the pattern is very similar to that obtained by Ginoux et al. (2012; their Fig.
5), although there are some differences, largely due to the masked DOD (i.e., from Step
1) used in this study and a lower threshold in less dusty regions.

Note that the selections of masking criteria in Step 1 and $DOD_{thresh}$ in Step 4 are

empirical and can add uncertainties to this method. Also, we approximate dust emission
using cumulative frequency of DOD, which may overestimate dust emission in regions
where the contribution of transported dust is significant and thus underestimate the
$V_{threshold}$ in those regions.



**2.3 Simulation design**


345 The AM4.0/LM4.0 is a coupled land-atmosphere model newly developed at

346 GFDL (Zhao et al., 2018a,b). It uses the recent version of the GFDL Finite-Volume

347 Cubed-Sphere dynamical core (FV$^3$; Putman and Lin, 2007), which is developed for

348 weather and climate applications with both hydrostatic and non-hydrostatic options.

349 Some substantial updates have been incorporated into the AM4.0, such as an updated

350 version of the model radiation transfer code, an alternate topographic gravity wave drag

351 formulation, a double-plume model representing shallow and deep convection, a "light"

352 chemistry mechanism, and modulation on aerosol wet removal by convection and frozen

353 precipitation (Zhao et al., 2018a,b). Here we used a model version with 33 vertical levels

354 (with model top at 1hPa) and cube-sphere with 192×192 grid boxes per cube face

355 (approximately 50 km grid size).

356 The aerosol physics is based in large part on that of the GFDL AM3.0 (Donner et

357 al., 2011), but with a simplified chemistry where ozone climatology from AM3.0

358 simulation (Naik et al., 2013) is prescribed. AM4.0 simulates the mass distribution of five

359 aerosols: sulfate, black carbon, organic carbon, dust, and sea salt. Dust is partitioned into

360 five size bins based on radius: 0.1~1 μm (bin 1), 1~2 μm (bin 2), 2~3 μm (bin 3), 3~6 μm

361 (bin 4), and 6~10 μm (bin 5). The dust emission scheme follows the parameterization of

362 Ginoux et al. (2001), as shown in the following equation:

$$F_p = C \times S \times s_p \times V_{10m}^2 (V_{10m} - V_t) \qquad (\text{if } V_{10m} > V_t), \qquad (3)$$


365 where $F_p$ is flux of dust of particle size class $p$, $C$ is a scaling factor with a unit of μg s$^2$

366 m$^{-5}$, here $C$ is set to $0.75 \times 10^{-9}$. $S$ is the source function based on topographic depressions



(Ginoux et al., 2001), $s_p$ is fraction of each size class, and $V_{10m}$ is surface 10 m wind
speed, and $V_t = 6$ m s$^{-1}$ is the threshold of wind erosion.

Three simulations with prescribed sea surface temperature (SST) and sea ice

(Table 2) were conducted from 1999 to 2015, with the first year discarded for spin up.
The Atmospheric Model Intercomparison Project (AMIP)-style SST and sea ice data
(Taylor et al., 2000) are from the Program for Climate Model Diagnosis and
Intercomparison (PCMDI), which combined HadISST (Rayner et al., 2003) from UK Met
Office before 1981 and NCEP Optimum Interpolation (OI) v2 SST (Reynolds et al.,
2002) afterwards. The surface winds in the simulations are nudged toward the NCEP1
reanalysis with a relaxation timescale of 6 hours (Moorthi and Suarez, 1992). Note that
the nudged surface winds are actually weaker than the surface wind speed simulated by
the standard version of AM4.0/LM4.0 without nudging, so the overall magnitude of dust
emission is lower than the standard version. Here we choose not to retune the dust
emission scheme but instead test the usage of $V_{threshold}$, which theoretically provides a
more physics-based way to improve dust simulation.

In the Control run, the default model setting is used for dust emission, with a

prescribed 6 m s$^{-1}$ threshold of wind erosion (cf. Ginoux et al., 2019). In the $V_{thresh}$12mn
simulation, the observation based climatological monthly $V_{threshold}$ is used to replace the
constant wind erosion threshold. The default source function $S$ in Eq. 3 only allows dust
emission over bare ground by masking out regions with vegetation cover. Since LAI
masking is already applied in the retrieval of $V_{threshold}$ (i.e., LAI<0.3), we choose to use a
source function that is the same as the default source function $S$ but without vegetation
masking, i.e., $S'$ (Figure S2 in the supplement). This allows the influence of the spatial



and temporal variations of $V_{threshold}$ to be fully examined. The combination of source
function $S'$ and $V_{threshold}$ also extends dust source from bare ground to sparsely vegetated
area as outlined by $V_{threshold}$, e.g., over central North America, central India, and part of
Australia, and can increase dust emission in these regions. The pattern of extended dust
source area largely resembles the vegetated dust source identified by Ginoux et al. (2012;
their Fig. 15b) and Kim et al. (2013; their Fig. 9). All the other settings are the same as
the Control run. The $V_{thresh}12mn$ simulation is the same as the $V_{thresh}12mn$ but uses the
annual mean of $V_{threshold}$ for each month. Since the same SST and sea ice are prescribed
for all simulations, the differences in simulated dynamic vegetation by LM4.0 among the
three simulations are actually very small and can be ignored (see Figures S3-4 in the
Supplement).

**3. Results**
**3.1 Threshold of wind erosion**
Figures 1f-j show the derived threshold of wind erosion for each season and
annual mean. The seasonal variations of wind erosion are largely due to the variations of
land surface features examined here, such as soil moisture, soil temperature, snow cover,
and vegetation coverage in each month. $V_{threshold}$ is generally lower in MAM and JJA
(SON and DJF) for Northern (Southern) Hemisphere dusty regions than in other seasons.
$V_{threshold}$ values are also lower in major dust source regions (i.e., regions with a high FoO
in Figs. 1a-e). Globally, the lowest $V_{threshold}$ values (~3-5 m s$^{-1}$) are located over North
Africa and the Middle East, while the highest values (>10 m s$^{-1}$) occur over northern
Eurasia.



Figure 2a shows the cumulative frequency of $V_{threshold}$ over the global land area for

each season and annual mean. The globally constant threshold 6 m s$^{-1}$ used in the GFDL

AM4.0/LM4.0 is actually above the 50% level for all seasons and annual mean,

indicating the default setting in model likely overestimates the threshold of wind erosion.

In fact, the 50% level of $V_{threshold}$ is around 4.5 m s$^{-1}$ for the annual mean and ranges from

4 m s$^{-1}$ in JJA to about 5 m s$^{-1}$ in SON and DJF.

The distributions of $V_{threshold}$ for annual mean (black bars) and dusty seasons

(color lines; MAM and JJA for the Northern Hemisphere and SON and DJF for the

Southern Hemisphere) for each dusty region (see Fig. 1f and Table 1 for locations) are

shown in Figs. 2b-j. In the Sahel, the annual mean $V_{threshold}$ peaks around 4 m s$^{-1}$ (Fig. 2b).

This magnitude is lower than indicated from previous studies based on station

observations in the region, e.g., Helgren and Prospero (1987) found the threshold velocity

over eight stations in Northwest Africa ranged from 6.5 to 13 m s$^{-1}$ during summer in

1974. Chomette et al. (1999) and Marsham et al. (2013) also reported higher wind

erosion thresholds around 6-9 m s$^{-1}$ at individual stations.  On the other hand, Cowie et al.

(2014) found that the annual threshold of wind erosion at the 25% level, i.e., when

surface condition is favorable for dust emission, can be lower than 6 m s$^{-1}$ at some sites in

the Sahel (their Fig. 5). Several factors may contribute to the discrepancies. First, studies

suggest that reanalysis datasets may underestimate surface wind speed in spring and for

monsoon days in Africa (e.g., Largeron et al., 2015), and therefore could lead to a lower

value of $V_{threshold}$ than that derived from station observations. In fact, Bergametti et al.

(2017) found even 3-hourly wind speed record at stations may miss short events with

high weed speed.  As mentioned earlier, using DOD frequency to approximate dust





emission may lead to an overestimation of dust emission over regions such as the
southern Sahel where transported dust is a large component and consequently an
underestimation of $V_{threshold}$. Different analysis time periods or methods to retrieve the
wind erosion threshold may also contribute to the differences.

The annual mean $V_{threshold}$ in the Sahara and Arabian Peninsula is a bit higher,

with mean values at 4.5 and 5.2 m s$^{-1}$, respectively (Figs. 2c-d). The $V_{threshold}$ over
northern China is even higher, with an annual mean of 7.9 m s$^{-1}$. This is consistent with
the results of Kurosaki and Mikami (2007), who found that under favorable land surface
conditions the threshold wind speed ranges from 4.4± 0.6 m s$^{-1}$ in Taklimakan Desert to
6.9± 1.2 m s$^{-1}$ over the Loess Plateau and around 9.8± 1.6 m s$^{-1}$ in the Gobi Desert. These
values are also consistent with Ginoux and Deroubaix (2017) who found that regional
mean wind erosion threshold over northern China ranges from 6.5 to 9.1 m s$^{-1}$. In India,
the $V_{threshold}$ peaks at about 4.5 m s$^{-1}$ and 6.5 m s$^{-1}$, respectively (Fig. 2f). The second peak
is probably related to anthropogenic dust sources over the central Indian subcontinent
(Ginoux et al., 2012). We also note that in the Northern Hemisphere, the $V_{threshold}$ in dusty
seasons is shifted towards lower values than the annual mean (blue and green lines in
Figs. 2b-g), but is similar to the annual mean in the Southern Hemisphere, indicating
stronger influences of surface variability in the Northern Hemisphere.

**3.2 $V_{threshold}$ in the GFDL AM4.0/LM4.0 model**

The derived $V_{threshold}$ is then implemented into the GFDL AM4.0/LM4.0 models.

In this section we analyze the model output using the default setting (Control), 12-month





(V$_{thresh}$12mn), and annual mean $V_{threshold}$ (V$_{thresh}$Ann) to see how $V_{threshold}$ may affect the
simulation of DOD, surface dust concentration, and dust event frequency in the model.

**3.2.1 Climatology of AOD and DOD**

In order to compare the model results with observations, we first show the

climatology of AERONET AOD and COD from 2003 to 2015. As shown in Figure 3,
annual mean global AOD is highest over Africa, the Arabian Peninsula, Indian
subcontinent, and Southeast Asia.  In the latter two regions, high sulfate concentrations
(e.g., Ginoux et al., 2006) and organic carbon from biomass burning in Southeast Asia
(e.g., Lin et al., 2014) contribute substantially to the total AOD.  The SDA COD shows
the optical depth due to coarse aerosols, which includes both dust and sea salt, and sea
salt over coastal regions or islands can be a major contributor.  Here, high values (>0.2)
are largely located over dusty regions such as North Africa, the Arabian Peninsula, and
northern India (Fig. 3b).

Figures 4a-b show the scatter plots of modeled AOD and COD in the Control run

versus AERONET AOD and COD, respectively. Here column-integrated extinction from
both dust and sea salt is used to calculated COD in the model. The relative differences
(%) between AM4.0 output and AERONET station data are also shown (Figs. 4c-d). The
percentage of DOD to total COD in the model is displayed at the bottom (Fig. 4e). The
simulated AOD is lower than that from the AERONET over North Africa, the Middle
East, and western India, largely due to low values of COD simulated in these regions
(Fig. 4d). Besides these regions, the COD over North America, South America, South
Africa, and northern Eurasia is also, for the most part, underestimated by the model. Dust



is the dominant contributor to the COD value over most of these low COD regions,
except over the central to eastern North America and central South America.

The underestimation of COD (and effectively DOD given its dominance in most

regions) was improved in the subsequent model run using a prescribed 12-month $V_{threshold}$.
Figure 5 shows the results from the $V_{thresh}12mn$ simulation. COD is better captured while
the AOD effectively moves from a negative to a slightly positive bias (Figs. 5a-d). Most
sites over North Africa and the Middle East show a relatively small difference with
AERONET COD (Fig. 5d). Over the Indian subcontinent, COD is overestimated, while
over North America excluding the east coast, northern Eurasia, and part of South
America, COD is also better captured than in the Control run.

These improvements are largely associated with a better simulation of DOD in the

"dust belt" (i.e., North Africa and the Middle East). Figure 6 shows the DOD at 550 nm
derived from AERONET AOD (see methodology for details) versus that from the
$V_{thresh}12mn$ simulation. Over most stations in the Sahel, Mediterranean coasts, and
central Middle East, the relative differences between modeled and observed DOD is
within ± 25%.

Figure 7 shows the regional averaged annual mean DOD over nine dusty regions

from MODIS and three simulations. The Control run largely underestimates DOD in all
regions, while the magnitude of DOD is better captured in the $V_{thresh}12mn$ and $V_{thresh}Ann$
simulations, although slightly overestimated in the Sahel and greatly overestimated over
Australia. In general, DOD simulated by the $V_{thresh}Ann$ run using a constant annual mean
$V_{threshold}$ is higher than that simulated by the $V_{thresh}12mn$ run, consistent with the higher
dust emission in the $V_{thresh}Ann$ run (Table S2 in the Supplement). Lack of soil moisture





constraint in the model, which is a very important element in capturing the variation of
DOD in Australia (Evans et al., 2016), may contribute to the large overestimation of
DOD in Australia.

**3.2.2 Climatology of surface dust concentration**

While DOD is a key parameter associated with the climate impact of dust, surface

dust concentration is an important factor affecting local air quality. Here we compare the
modeled surface dust concentration with RSMAS station observations. Model output is
averaged from 2000 to 2015 to form the annual climatology. Consistent with the DOD
output, the Control run largely underestimates surface dust concentrations at almost all of
the sites (except sites 9 and 15; Figure 8 top panel). The underestimation bias is reduced
in the $V_{thresh}Ann$ simulation (Fig. 8, middle panel), with seven stations having
model/observation ratios between 0.5 and 2 (white triangles). Over the coastal U.S. (e.g.,
sites 16 and 13), dust concentrations are overestimated, consistent with the
overestimation of DOD over the U.S. and the Sahel (Fig. 7).  Dust concentrations in
Australia and the east coast of China are also overestimated by more than five-folds.
Surface dust concentration is further improved in the $V_{thresh}12mn$ simulation (Fig. 8,
bottom), with eight stations showing a model/observation ratio between 0.5 and 2 and
only four stations overestimating or underestimating dust concentrations by more than
five times.

Simulated surface fine dust concentration (calculated as dust bin 1+0.25×dust bin

2) in the U.S. is compared with gridded IMPROVE data (Figure 9). While the Control
run largely underestimates surface fine dust concentration, the simulated concentration is



overall too high in the $V_{thresh}$Ann run. The spatial pattern of fine dust concentration is
better captured in the $V_{thresh}$12mn run, with higher values over the southwestern U.S., but
the magnitude is still overestimated, and additional dust hot spots are simulated over the
northern Great plains and the Midwest, which are not shown in the IMPROVE data. Such
an overall overestimation may be attributed to lack of soil moisture modulation in the
dust emission scheme. The way in which dust bins are partitioned in the model can add
uncertainties to model's representation of surface fine dust concentrations as well.  On
the other hand, the relatively low spatial coverage of IMPROVE sites over the northern
Great Plains and Midwest (e.g., Pu and Ginoux, 2018a) may also add uncertainties to the
data itself.

**3.2.3 Seasonal cycles**

Figure 10 compares the seasonal cycle of DOD from three simulations with

MODIS DOD in nine dusty regions. The seasonal cycle of gridded AERONET COD (as
an approximation to DOD; on a 0.5° by 0.5° grid) is also shown. Since the gridded COD
may have large uncertainties over regions with only a few stations, such as the Sahel,
Sahara, northern China, and South Africa, MODIS DOD is used as the main reference in
the comparison. Seasonal cycles are better captured by the $V_{thresh}$12mn simulation in the
Sahel, the Sahara, and the Arabian Peninsula (Figs. 10a-c), although the spring and
summer peak in the Sahel is overestimated and winter minimum in the Sahara is
underestimated.  The MAM peak of MODIS DOD in northern China is missed by both
$V_{thresh}$12mn and $V_{thresh}$Ann simulations (Fig. 10d), while the JJA peak over India is
largely overestimated (Fig. 10e). Over the U.S. dusty region, the seasonal cycle in the



$V_{thresh}12mn$ simulation is slightly underestimated compared to MODIS DOD but
overestimated from May to August in the $V_{thresh}$Ann simulation (Fig. 10f). DOD is
underestimated in South Africa in all three simulations (Fig. 10g). Over South America,
the peak from October to February is roughly captured by the $V_{thresh}12mn$ run but is
overestimated by the $V_{thresh}$Ann run (Fig. 10h). The seasonal cycles of DOD in Australia
are very similar in all three simulations and largely resemble that in the MODIS, although
both the $V_{thresh}12mn$ and $V_{thresh}$Ann simulations overestimate the DOD by about an order
of magnitude.

Figure 11 shows the seasonal cycle of COD from 12 AERONET SDA sites over

North Africa and nearby islands (see Figure S5 in the Supplement for site locations)
along with MODIS DOD and DOD simulated in three runs. The magnitude of
AERONET COD and MODIS DOD in these sites are very similar, despite missing values
at sites 1, 4, 5, 8, 11, and a smaller value at site 2 in MODIS. Over most of the sites, the
seasonal cycle is better captured in the $V_{thresh}12mn$ and $V_{thresh}$Ann simulations than the
Control run, although the peak over Cairo_EMA_2 (site 12) is slightly underestimated,
which is consistent with the underestimation of annual mean DOD in the area (Fig. 6).

We also examined the seasonal cycle of PM10 surface concentration at three

Sahelian INDAAF stations (see Figure S5 in the Supplement for site locations) from the
LISA project. Figures 12a-c show $PM_{10}$ surface dust concentration (here dust dominates
total $PM_{10}$ concentration) from the Control, $V_{thresh}12mn$, and $V_{thresh}$Ann simulations
versus observed $PM_{10}$ concentration from three LISA sites. $PM_{10}$ concentrations in these
sites peak during boreal winter and spring and reach minima from July to September.
These seasonal variations are associated with the dry northerly Harmattan wind in boreal



winter that transports Saharan dust southward to the Guinean coast and the scavenging
effect of monsoonal rainfall in boreal summer that removes surface dust (Marticorena et
al., 2010). While the Control run does not capture the seasonal cycles in these sites, the
$V_{thresh}12mn$ run largely captures the spring peak and summer minimum, although the
magnitude is overestimated. In all three sites, the simulated concentration in the
$V_{thresh}Ann$ run is larger than that in the $V_{thresh}12mn$ run, especially in boreal fall to early
spring. Such an overestimation is probably due to the prescribed constant annual mean
$V_{threshold}$, which is lower than it would be during the less dusty season (i.e., boreal fall to
winter) and thus increases dust emission and surface concentration.

Figs. 12d-f show the seasonal cycle of DOD from three AERONET sites co-

located with LISA INDAAF stations and from three simulations. The $V_{thresh}12mn$ and
$V_{thresh}Ann$ simulations largely captured the seasonal cycle of DOD at these sites. The
overestimation of near surface $PM_{10}$ dust concentration (Figs. 12a-c) and the generally
well-captured column integrated DOD (Figs. 12d-f) indicate that model likely
underestimates dust concentration in the atmospheric column above the surface, which
needs further investigation in future studies.

**3.2.4 A dust storm over U.S. northern Great Plains on October 18[th], 2012**

Can the AM4.0/LM4.0 with prescribed $V_{threshold}$ better represent individual dust

events? Here we examine a major dust storm captured by MODIS Aqua true color-image
on Oct. 18[th], 2012 (https://earthobservatory.nasa.gov/images/79459/dust-storm-in-the-
great-plains) over the U.S. northern Great Plains. There was a severe drought in 2012
with anomalously low precipitation centered over the central U.S. (e.g., Hoerling et al.,





2014).  The dry conditions favored dust storm development when there were intensified
surface winds.  However, this storm was not predicted by the forecast models, such as the
Goddard Earth Observing System version 5 (GEOS-5; Rienecker et al., 2008) and Navy
Aerosol Analysis Prediction System (NAAPS; Witek et al., 2007; Reid et al., 2009;
Westphal et al., 2009).

As shown in Figure 13, MODIS DOD also captures this event, with a peak value

above 0.5 over southwest Nebraska and northern Kansas on Oct. 18$^{th}$, 2012. The
$V_{thresh}$12mn run also largely captures this event (Fig. 13 bottom panel), although the
Control run totally misses it (not shown). In the model, the dust storm appears in South
Dakota and Nebraska on Oct. 17$^{th}$, 2012, along with the anomalous southwesterly winds.
It reaches a maximum on Oct. 18$^{th}$, in association with intensified anomalous
southwesterly winds at the surface and an anomalous low-pressure system at 850 hPa
(Figure S6 in the Supplement).   Note that the modeled dust storm centers a bit
northeastward compared to the MODIS DOD pattern and it also has greater magnitude
and covers a larger area. On Oct. 19$^{th}$, both the anomalous low-pressure system and
surface wind speeds weaken and the dust storm dissipates, with slightly elevated DOD
levels over a region extending over the lower Mississippi River basin and the Midwest.
This is somewhat consistent with MODIS records, which also shows slightly higher DOD
levels over Tennessee and northern Alabama on Oct. 19$^{th}$, regardless of large area of
missing values.

**3.3 Frequency of DOD in the model versus that from MODIS**




Figure 14 shows the frequency of regional mean DOD during one dusty season

(MAM in the Northern Hemisphere and SON in the Southern Hemisphere) for nine

regions. Results from MODIS, the Control, and $V_{thresh}$12mn runs are shown in black,

blue, and orange lines, respectively. In most dusty regions, such as the Sahara, Sahel,

Arabian Peninsula, India, and northern China, MODIS DOD frequency largely peaks

between 0.2 to 0.4, while DOD frequency peaks at a much lower level between 0.02 to

0.08 in less dusty regions, such as the U.S., South America, South Africa and Australia.

The DOD distribution in the Control run is biased low and peaks around 0.05 in those

dusty regions and between 0 and 0.01 in less dusty regions. The frequency is much better

captured in the $V_{thresh}$12mn run over the Arabian Peninsula and the Sahel, slightly

improved but still biased low over the Sahara, northern China, India, and the U.S. The

modeled frequency in the $V_{thresh}$12mn run is biased high in Australia (peaks outside the

maximum of x-axis, not shown) and shows little improvement over South Africa and

South America. The overall improvement of DOD frequency using the time-varying 2D

$V_{threshold}$ occurs mostly over major dusty regions, which is consistent with the

improvements in DOD climatology and seasonal cycle in the model simulations.

**4. Discussion**

A global distribution of the threshold of wind erosion is retrieved using high

resolution MODIS DOD and land surface constraints from relatively high–resolution

satellite products and reanalyses. While this climatological monthly $V_{threshold}$ provides

useful information about the spatial and temporal variations of wind erosion threshold,

there are some uncertainties associated with it. Here DOD frequency is derived using



MODIS and other satellite products, thus the uncertainties in the satellite products are
inherited in the derived DOD frequency distribution. Due to the cloud screening
processes of MODIS products, dust activities over cloud-covered regions may be
underestimated. Also, DOD frequency is derived based on daily observations over a 13-
year record, so that some variability of dust emission associated with alluvial sediments
deposited by seasonal flooding may be not captured. Diurnal variability of dust emission
and short-duration events such as haboobs are also not included.  Since DOD is a column
integrated variable, it includes both local emitted and remotely transported dust. When
using DOD frequency distribution to approximate dust emission, it may overestimate dust
emission in regions where transported dust is dominated, e.g., over the southern Sahel,
and lead to an underestimation of $V_{threshold}$.
Previous study found that over regions such as North Africa, reanalysis products
may underestimate surface wind speed in spring and monsoon seasons but overestimate it
during dry nights (e.g., Largeron et al., 2015). This is largely because mechanisms such
as density current that can enhance surface wind speed are not parameterized in the
atmospheric models to produce the reanalysis products, while coarse spatial and temporal
sampling may also contribute to the underestimation of reanalysis wind speeds. These
limitations add uncertainties to the $V_{threshold}$ estimates derived here.
In addition, $V_{threshold}$ is derived by matching the frequency distribution of DOD at
certain levels (0.2 or 0.02) with the frequency distribution of daily maximum wind, and
these two values are derived empirically. The influences of soil properties such as soil
cohesion, particle size, and particle compositions on the threshold of wind erosion (e.g.,





Fécan et al., 1999; Alfaro and Gomes, 2001; Shao, 2001; Kok et al., 2014b) are not
explicitly examined here and will need further investigation.

The influences of $V_{threshold}$ on AM4.0/LM4.0 results are twofold. On the one hand,

it modifies the default constant threshold of wind erosion ($V_t$ in Eq. 3) by allowing spatial
and temporal variations of wind erosion threshold over bare ground, i.e., within the
domain of default dust source function $S$ (Figs. S7a-e in the Supplement). On the other
hand, it slightly extends the potential emission area to sparsely-vegetated regions as
outlined by $V_{threshold}$ (Figs. S7f-j in the Supplement). Which effect dominates? Taking the
$V_{thresh}$12mn simulation as an example, Figure S8 shows the differences of dust emission
with the Control run. The increase of dust emission in the $V_{thresh}$12mn simulation (also
summarized in Table S2 in the Supplement) is largely associated with the enhanced
emission over the bare ground (Figs. S8a-e in the Supplement), mainly over the regions
with reduced wind erosion threshold (Figs. S7a-e in the Supplement). The increased
emission over sparsely-vegetated area over regions such as the southern Sahel, India, and
Australia plays a minor role. This is consistent with Kim et al. (2013), who found global
dust emission in the Georgia Institute of Technology–Goddard Ozone Chemistry Aerosol
Radiation and Transport (GOCART) model is dominated by emission from bare ground.

**5. Conclusion**

While dust aerosols play important roles in the Earth's climate system, large

uncertainties exist in modeling its lifecycle (e.g., Huneeus et al., 2011; Pu and Ginoux,
2018b). Constant thresholds of wind erosion are widely used in climate models for
simplicity. Here, high-resolution MODIS Deep Blue dust optical depth (DOD) and



surface wind speeds from the NCEP1 reanalysis, along with other land surface factors
that affect wind erosion, such as soil moisture, vegetation cover, snow cover, soil
temperature, and soil depth, were used to develop a time-varying two-dimensional
climatological threshold of wind erosion, $V_{threshold}$, based on the seasonal variations of
DOD and surface wind distribution frequencies. $V_{threshold}$ is generally lower in dusty
seasons, i.e., MAM and JJA (SON and DJF) in the Northern (Southern) Hemisphere.
Globally, the lowest $V_{threshold}$ (~3-5 m s$^{-1}$) is located over North Africa and the Arabian
Peninsula, with the highest values (>10 m s$^{-1}$) over northern Eurasia.

The climatological monthly $V_{threshold}$ was then incorporated into the GFDL

AM4.0/LM4.0 model to examine the potential benefits relative to the use of a constant
threshold. In comparison with the simulation using the default setting of a globally
constant threshold of wind erosion (6 m s$^{-1}$), the frequency distribution, magnitude, and
seasonal cycle of DOD are largely improved over Northern Hemisphere dusty regions,
such as North Africa and the Arabian Peninsula, and slightly improved over India, the
western to central U.S., and northern China. The magnitude and seasonal cycle of DOD
are also slightly improved in South America, although change little in South Africa. The
incorporation of $V_{threshold}$ leads to an overestimation of DOD in Australia, likely in
association with the absence of soil moisture constraints on dust emission in the model.

The overall underestimation of surface dust concentration under default model

setting is largely reduced when time-varying $V_{threshold}$ is incorporated, except over a
central Pacific island and a Icelandic island where the concentration is still
underestimated and over Australia and coastal China where dust concentration is
overestimated. The spatial pattern of surface fine dust concentration in the U.S. is also



better captured, with the maximum of annual mean largely located over the southwestern
U.S., although the magnitude is overestimated.

A constant annual mean $V_{threshold}$ is also tested in the model, and is found to

overestimate DOD over dusty seasons in the Arabian Peninsula, U.S., India, Australia,
and South America.  Surface $PM_{10}$ concentrations in the Sahel during boreal fall and
winter seasons are also largely overestimated with this setting. The results indicate the
importance of including the seasonal cycle of $V_{threshold}$ in the model. Using time-varying
$V_{threshold}$, the model was also able to capture a strong dust storm in the U.S. Great Plains
in October 2012, which created deadly accidents, while some dust forecasting models
failed to reproduce it.

Finally, this method to retrieve global threshold of wind erosion can be

conducted under different resolutions or surface wind reanalsyses to match the resolution
of dust models and may help improve their simulations and forecasting of dust
distribution.











*Data availability.* Both the monthly and annual mean $V_{threshold}$ data at a 0.5° by 0.5°
resolution in NetCDF format is archived at: https://www.gfdl.noaa.gov/pag-
homepage/

*Author contributions*. PG and BP conceived the study. PG processed the MODIS Deep
Blue aerosol data and guided model simulations. HG, SM, VN, ES, and MZ assisted with
model configurations, while CH, JK, BM, NO, CG, and JP provided guidance on data
usage and analysis. BP conducted model simulations, analyzed data and model results,
and wrote the paper with contributions from all other co-authors.

*Acknowledgements.*

This research is supported by NOAA and Princeton University's Cooperative

Institute for Climate Science and NASA under grant NNH14ZDA001N-ACMAP and
NNH16ZDA001N-MAP. The authors thank Drs. Veronica Chan and Hyeyum Shin for
their helpful comments on the early version of this paper and Dr. Sophie Vandenbussche
for her valuable suggestions. We also thank the AERONET program for establishing and
maintaining the sunphotometer sites used in this study and the IMPROVE network for
the data. IMPROVE is a collaborative association of state, tribal, and federal agencies
and international partners. The US Environmental Protection Agency is the primary
funding source, with contracting and research support from the National Park Service.
The Air Quality Group at the University of California, Davis is the central analytical
laboratory, with ion analysis provided by Research Triangle Institute, and carbon analysis
provided by Desert Research Institute.

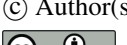



The AERONET aerosol optical depth data and SDA data are downloaded from

https://aeronet.gsfc.nasa.gov/new_web/download_all_v3_aod.html (last access: June
2018; Holben et al. 1998). IMPROVE fine dust data are downloaded from
http://views.cira.colostate.edu/fed/DataWizard/ (last access: March 2017, Malm et al.,
1994; Hand et al., 2011).






















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



Table 1 Major dusty regions shown in Figure 1. Note that region names such as India and
northern China are not exactly the same as their geographical definitions but also cover
some areas from nearby countries.

Table 2 Simulation design




















Figure 1. (a)-(e) Frequencies of occurrence (FoO; unit: days per season) in each season
and annual mean. (f)-(j) Threshold of wind erosion ($V_{threshold}$; unit: m s$^{-1}$) derived from
satellite products and reanalyses for each season and annual mean. Black boxes in (f)
denote nine dusty regions as listed in Table 1.

Figure 2. (a) Cumulative frequency of $V_{threshold}$ over global land for each season (black,
orange, blue, green, and grey lines denote annual, SON, JJA, MAM, and DJF averages,
respectively). Color dashed lines correspond to the percentages of $V_{threshold} = 6$ m s$^{-1}$ for
each season and annual mean. Color arrows point to the value of $V_{threhsold}$ at the 50% level
in each season and annual mean. (b)-(i) distribution of annual mean $V_{threshold}$ (black bars)
in each region (black boxes in Fig. 1) and for dusty seasons, i.e., MAM (green) and JJA
(blue) for regions in the Northern Hemisphere and SON (orange) and DJF (grey) for
regions in the Southern Hemisphere. The mean and ± one standard deviations of $V_{threshold}$
in each region are shown on the top right of each plot.

Figure 3. Climatology of annual mean AERONET (a) AOD (550 nm) and (b) SDA COD
(500 nm) averaged over 2003-2015.

Figure 4. Scatter plot of simulated annual mean (a) AOD and (b) COD in the Control run
versus AERONET AOD and COD (left), and the relative difference (in percentage) (c)
between modeled AOD and AERONET AOD and (d) between modeled COD and
AERONET COD (right). (e) The relative contribution of DOD to COD in the model.





Figure 5. Same as Fig. 4 but for the $V_{thresh}$12mn simulation.

Figure 6. (a) Climatology (2003-2015) of AERONET DOD (550 nm) over major dusty
regions and (b) scatter plot of modeled DOD in the $V_{thresh}$12mn simulation versus
AERONET DOD, and (c) the relative difference (in percentage) between modeled DOD
and AERONET DOD in the $V_{thresh}$12mn simulation.

Figure 7. Regional averaged annual mean DOD (2003-2015) over nine regions from the
Control (grey), $V_{thresh}$12mn (orange), and $V_{thresh}$Ann (yellow) simulations and MODIS
(black).

Figure 8. Scatter plots (left column) of model simulated (from top to bottom are the
Control, $V_{thresh}$Ann, and $V_{thresh}$12mn simulations) surface dust concentration versus the
climatology of observed surface dust concentration from RSMAS stations (Savoie and
Prospero 1989), and spatial pattern of surface dust concentration from model output
(shading; right column) and the ratio between modeled and RSMAS station observed
surface dust concentration (color triangles, with upward triangles indicating
overestimation and downward triangles indicating underestimation). 16 stations were
used, and numbers in each triangle (right) and grey dots (left) indicate the stations. The
one-one, one-two and one-five lines are plotted in solid, dashed and dash-dotted lines in
the scatter plots.



Figure 9. Annual mean surface fine dust concentration (μg m$^{-3}$) from IMPROVE stations
(left column) and three simulations (middle column) and the differences between model
and observation (right column) for 2002-2015.

Figure 10. Seasonal cycle of DOD from MODIS (black), the Control (grey), $V_{thresh}12mn$
(orange), and $V_{thresh}Ann$ (yellow) runs, and gridded AERONET SDA COD (blue)
averaged over nine regions. The annual mean of each dataset in each region is listed on
the top of the plot.

Figure 11. Seasonal cycle of DOD over 12 AERONET SDA sites (see Fig. S5 in the
Supplement for locations) from the Control (grey), $V_{thresh}12mn$ (orange), and $V_{thresh}Ann$
(yellow) simulations, along with DOD from MODIS (blue), and COD from AERONET
(black dotted line). All values are averaged over 2003-2015. The location (lat/long) and
the name (due to space, only first seven characters are shown) of the sites are listed at the
top of each plot.

Figure 12. (a)-(c) Seasonal cycle of PM$_{10}$ surface concentration (black) over three sites
from the LISA project, along with PM$_{10}$ surface dust concentration from the Control
(grey), $V_{thresh}12mn$ (orange), and $V_{thresh}Ann$ (yellow) simulations. Error bars are ± one
standard deviations of daily mean in each month averaged over 2006-2014. Unites: μg m$^{-}$
$^{3}$. (d)-(f) seasonal cycle of DOD (550 nm) from three AERONET sites co-located with
LISA sites (blue) versus that modeled by the Control (grey), $V_{thresh}12mn$ (orange), and
$V_{thresh}Ann$ (yellow) simulations.





Figure 13. Daily DOD from MODIS (top panel), daily DOD simulated by the $V_{thresh}$12mn
run along with anomalies (with reference to the 2000-2015 mean) of surface wind vectors
(m s$^{-1}$; bottom panel) from Oct. 17$^{th}$ to Oct. 19$^{th}$, 2012. Only DOD over land is shown.
Missing values in MODIS DOD (top panel) are plotted in grey shading.

Figure 14. Frequency (%) distribution of regional averaged daily DOD from MODIS
(black) versus that from the Control (light blue) and $V_{thresh}$12mn (orange) simulations for
the Sahara, the Sahel, the Arabian Peninsula, northern China, India, western to central
U.S., South America, South Africa, and Australia from 2003 to 2015. X-axis denotes the
ranges of DOD, and y-axis is percentage of occurrence. The light green boxes denote the
averaging areas. For regions in the Northern Hemisphere frequency in MAM is shown,
while for regions in the Southern Hemisphere frequency in SON is shown.













Table 1 Major dusty regions shown in Figure 1. Note that region names such as India and
northern China are not exactly the same as their geographical definitions but also cover
some areas from nearby countries.

| No. | Regions | Lat/long |
|---|---|---|
| 1 | Sahel | 10°-20°N, 18°W-35°E |
| 2 | Sahara | 20°-35°N, 15°W-25°E |
| 3 | Arabian Peninsula | 15°-35°N, 35°-60°E |
| 4 | Northern China (N. China) | 35°-45°N, 77°-103°E |
| 5 | India | 20°-35°N, 60°-85°E |
| 6 | U.S. | 25°-45°N, 102°-125°W |
| 7 | South Africa (S. Africa) | 17°-35°S, 15°-30°E |
| 8 | South America (S. America) | 18°-55°S, 65°-75°W |
| 9 | Australia | 15°-35°S, 128-147°E |

Table 2 Simulation design

| Simulations | Wind erosion threshold | Source function |
|---|---|---|
| Control | $6 \text{ m s}^{-1}$ | $S$ |
| $V_{thresh}12mn$ | 12-month $V_{threshold}$ | $S'$ |
| $V_{thresh}Ann$ | Annual mean $V_{threshold}$ | $S'$ |




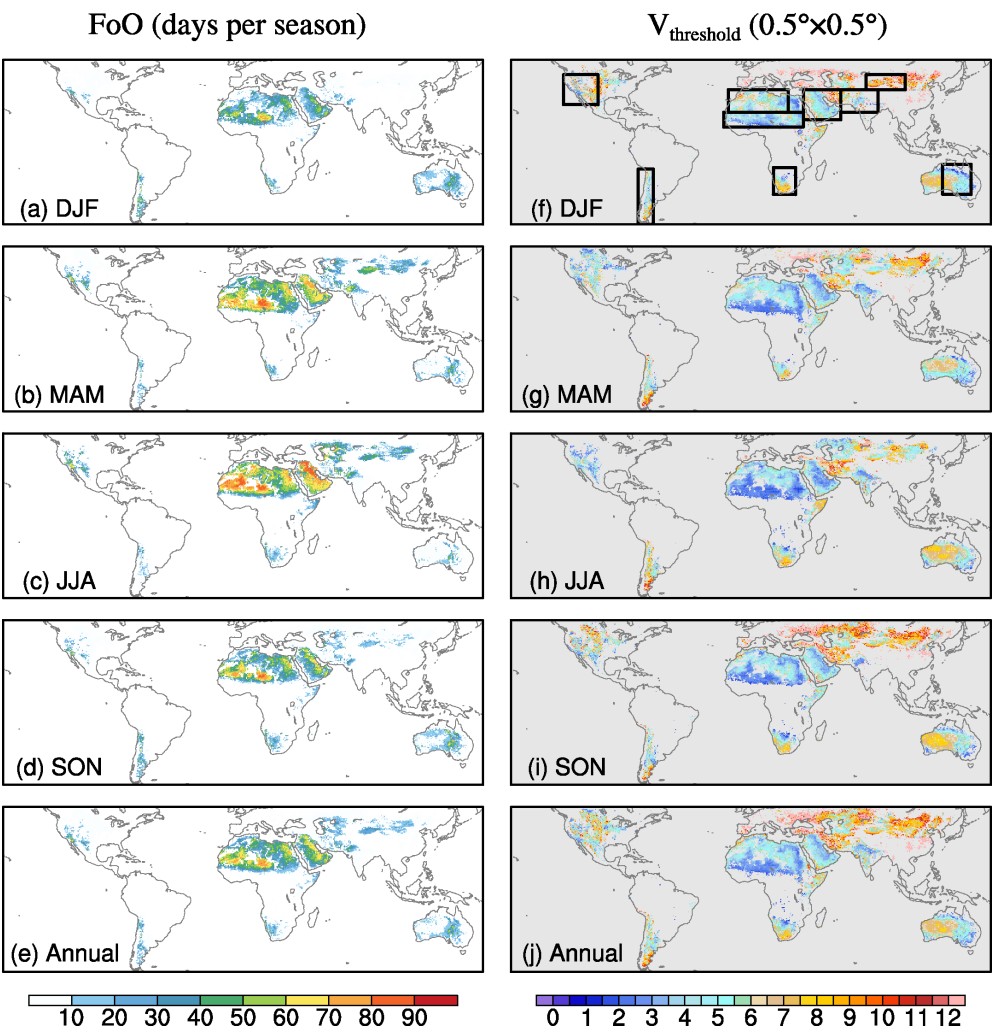

Figure 1. (a)-(e) Frequencies of occurrence (FoO; unit: days per season) in each season
and annual mean. (f)-(j) Threshold of wind erosion ($V_{threshold}$; unit: m s$^{-1}$) derived from
satellite products and reanalyses for each season and annual mean. Black boxes in (f)
denote nine dusty regions as listed in Table 1.

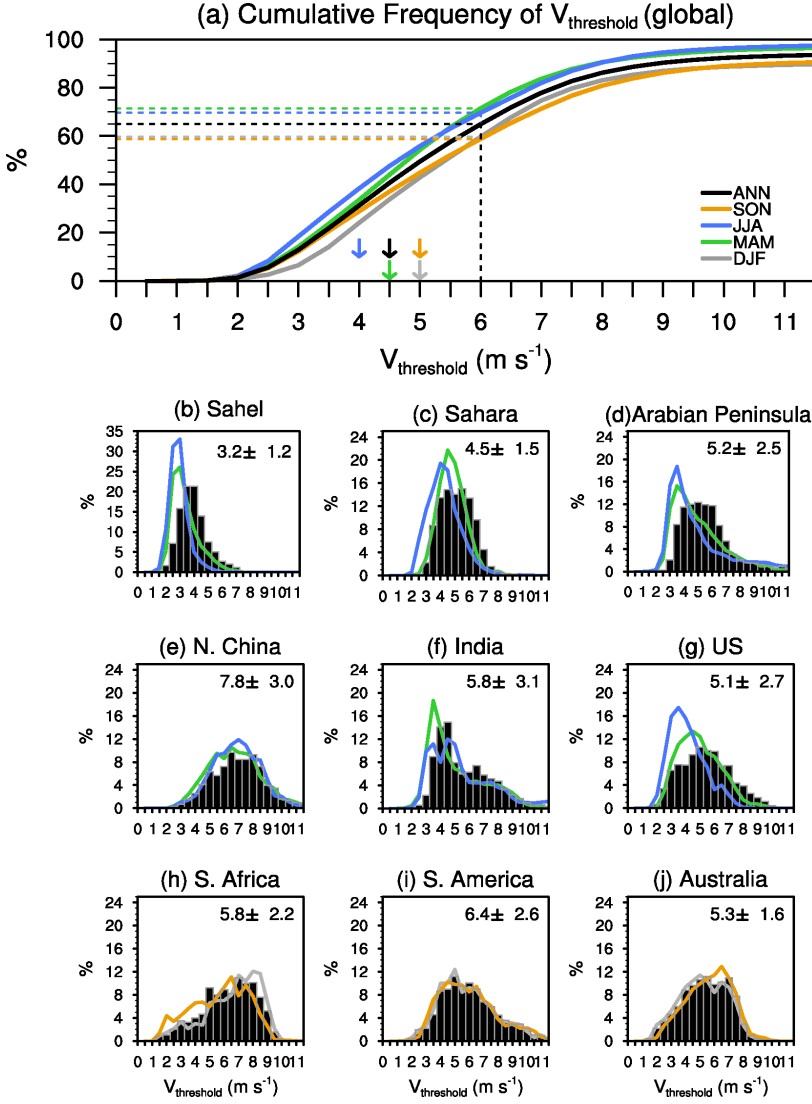

Figure 2. (a) Cumulative frequency of $V_{threshold}$ over global land for each season (black,
orange, blue, green, and grey lines denote annual, SON, JJA, MAM, and DJF averages,
respectively). Color dashed lines correspond to the percentages of $V_{threshold} = 6$ m s⁻¹ for
each season and annual mean. Color arrows point to the value of $V_{threhsold}$ at the 50% level
in each season and annual mean. (b)-(i) distribution of annual mean $V_{threshold}$ (black bars)
in each region (black boxes in Fig. 1) and for dusty seasons, i.e., MAM (green) and JJA
(blue) for regions in the Northern Hemisphere and SON (orange) and DJF (grey) for
regions in the Southern Hemisphere. The mean and ± one standard deviations of $V_{threshold}$
in each region are shown on the top right of each plot.



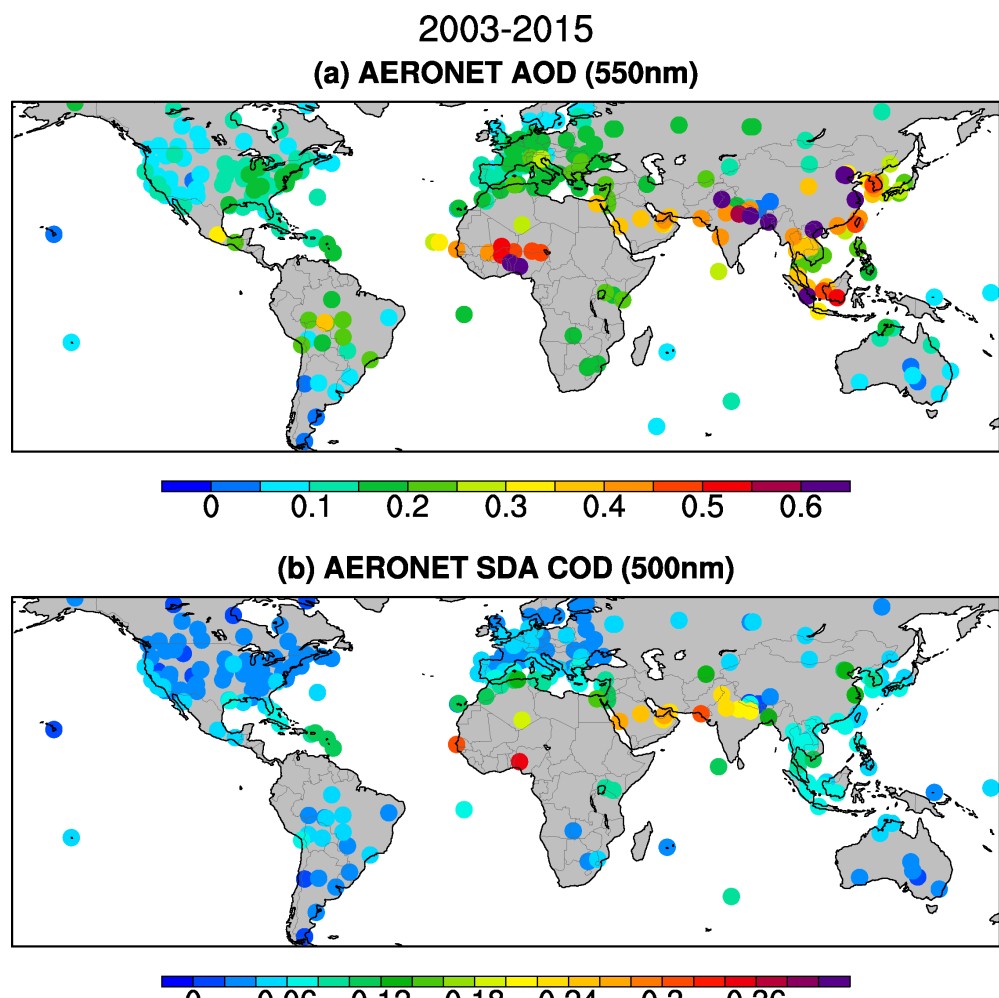

Figure 3. Climatology of annual mean AERONET (a) AOD (550 nm) and (b) SDA COD
(500 nm) averaged over 2003-2015.




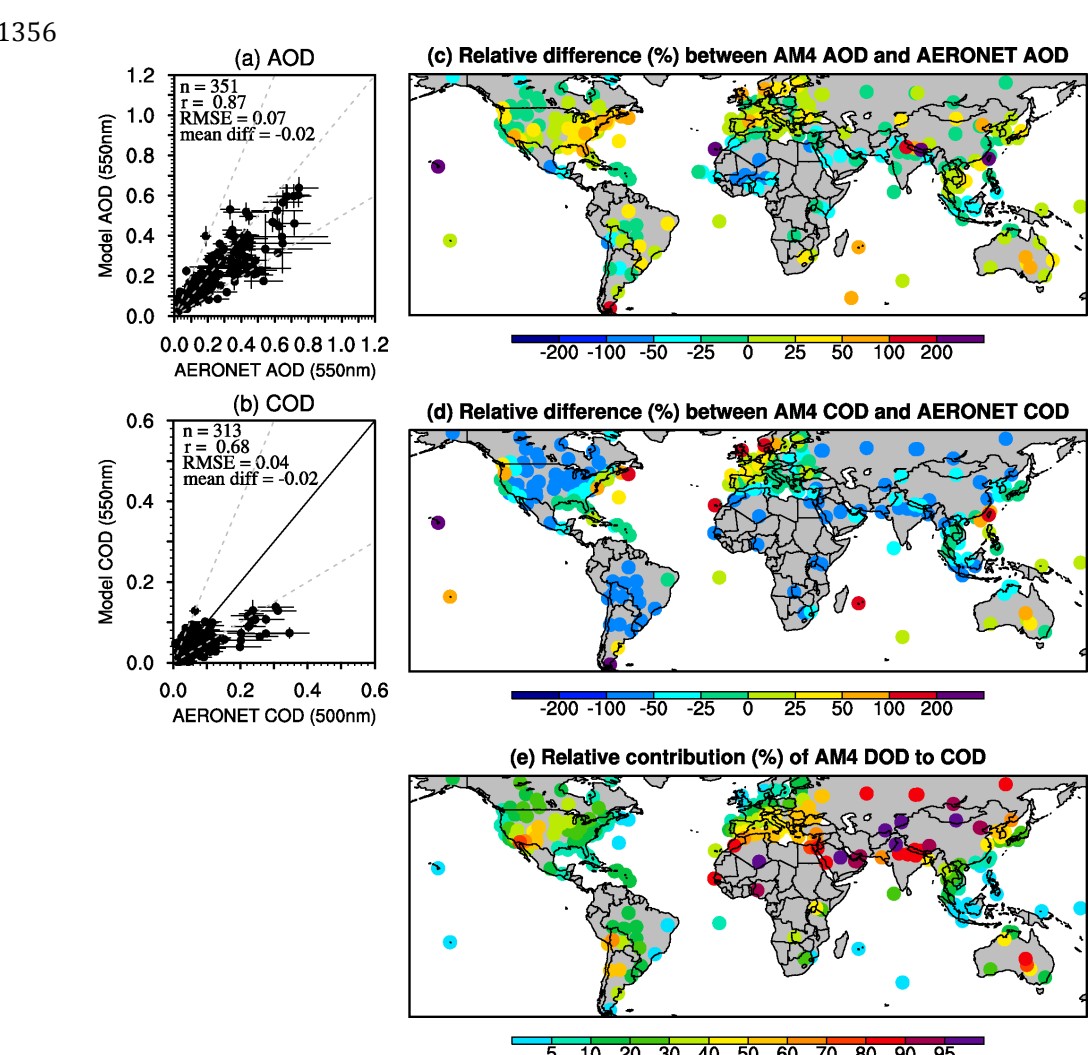

Figure 4. Scatter plot of simulated annual mean (a) AOD and (b) COD in the Control run
versus AERONET AOD and COD (left), and the relative difference (in percentage) (c)
between modeled AOD and AERONET AOD and (d) between modeled COD and
AERONET COD (right). (e) The relative contribution of DOD to COD in the model.








Figure 5. Same as Fig. 4 but for the $V_{thresh}12mn$ simulation.



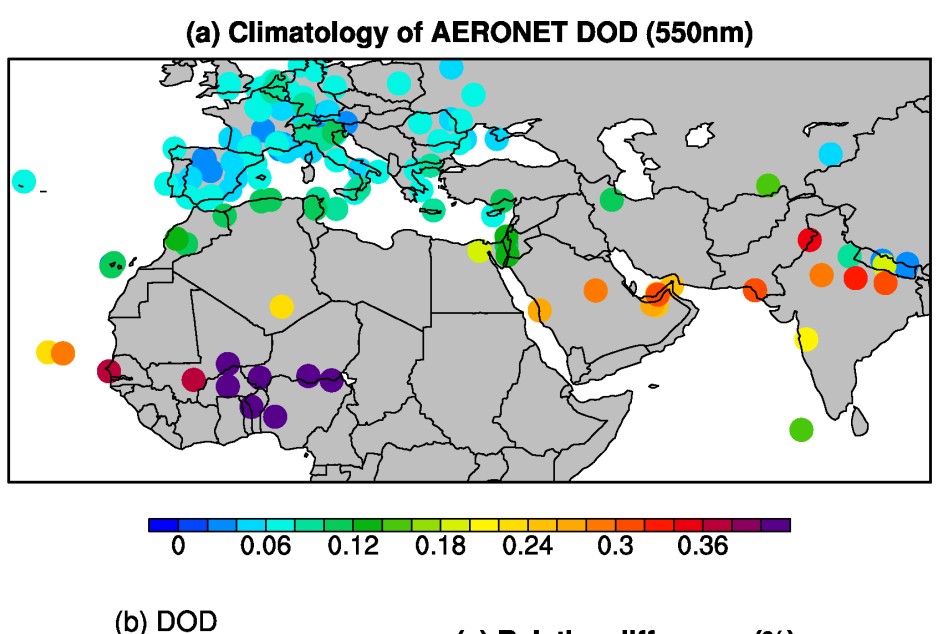

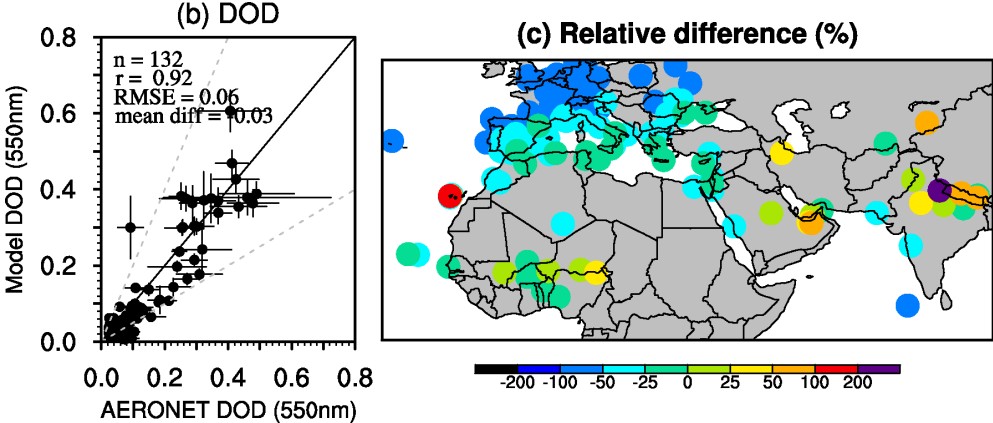

Figure 6. (a) Climatology (2003-2015) of AERONET DOD (550 nm) over major dusty
regions and (b) scatter plot of modeled DOD in the $V_{thresh}$12mn simulation versus
AERONET DOD, and (c) the relative difference (in percentage) between modeled DOD
and AERONET DOD in the $V_{thresh}$12mn simulation.




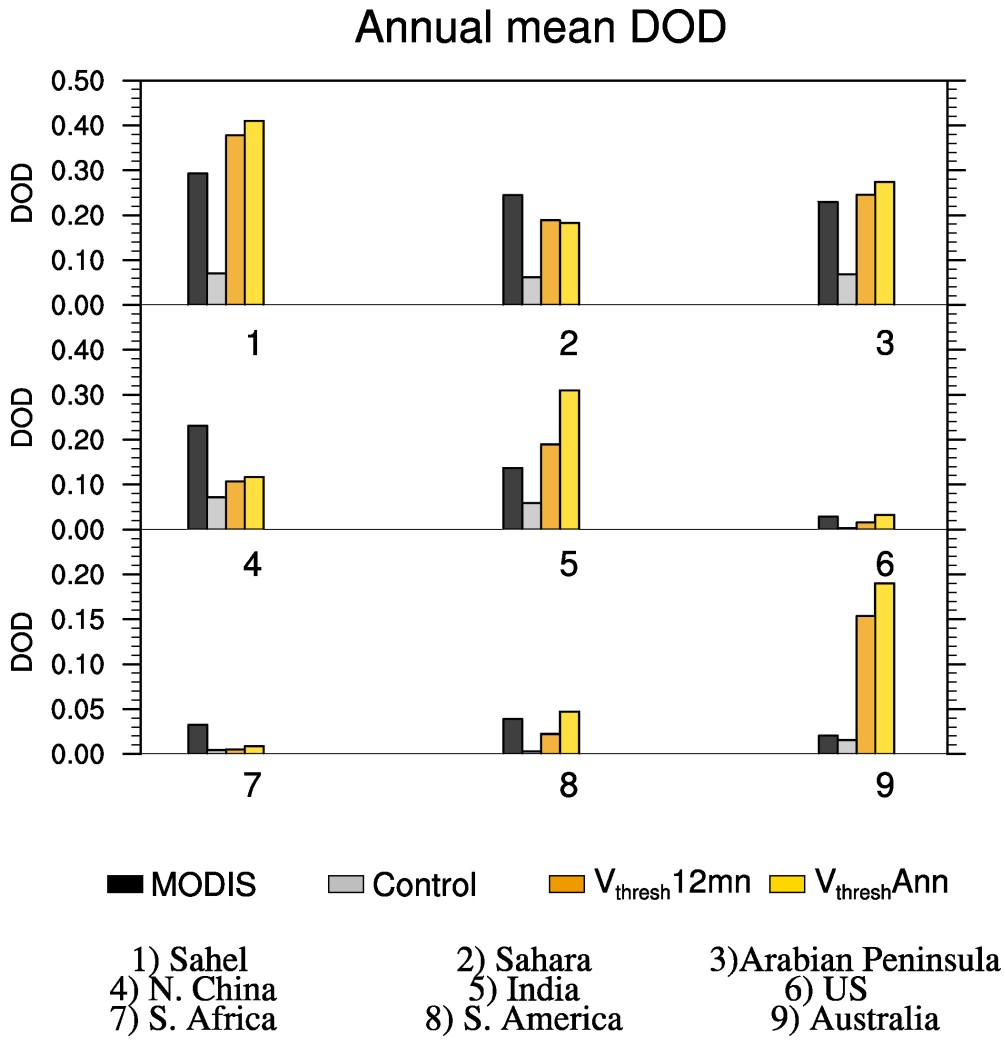

Figure 7. Regional averaged annual mean DOD (2003-2015) over nine regions from the
Control (grey), $V_{thresh}$12mn (orange), and $V_{thresh}$Ann (yellow) simulations and MODIS
(black).

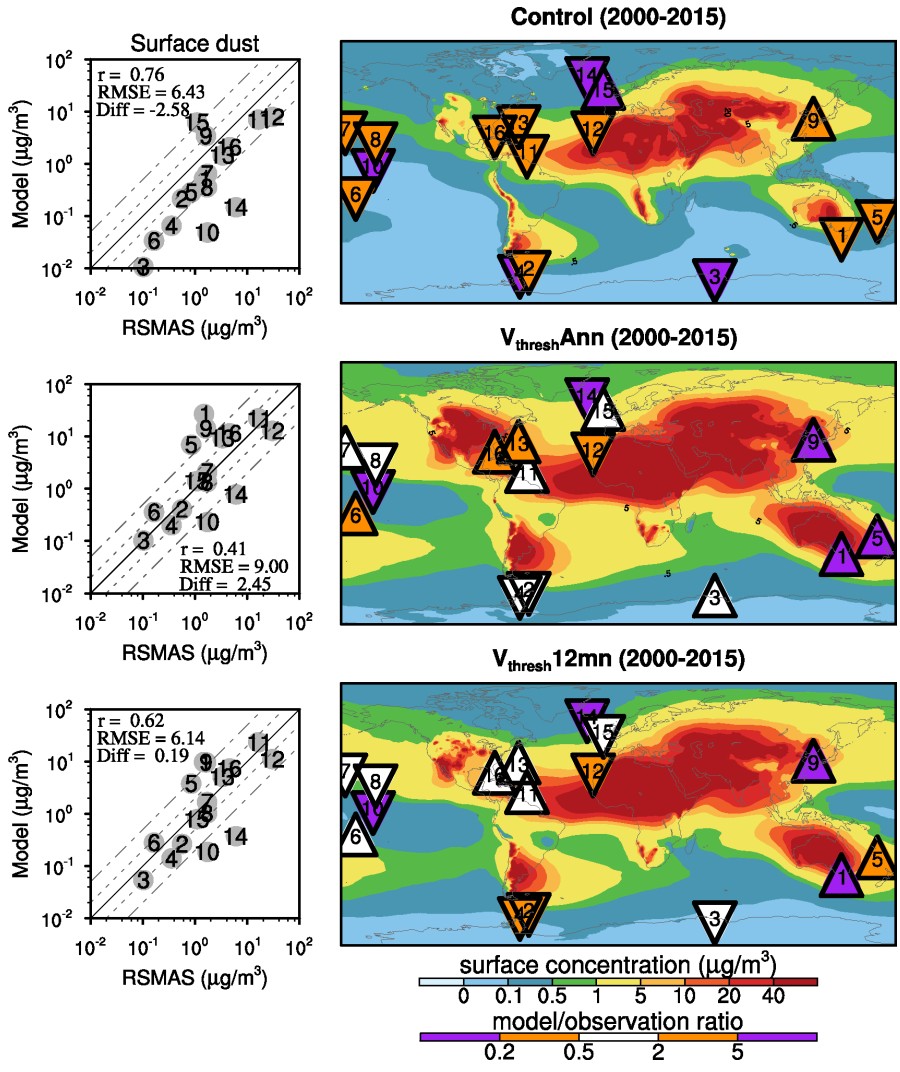

Figure 8. Scatter plots (left column) of model simulated (from top to bottom are the
Control, $V_{thresh}$Ann, and $V_{thresh}$12mn simulations) surface dust concentration (µg m$^{-3}$)
versus the climatology of observed surface dust concentration from RSMAS stations
(Savoie and Prospero 1989), and spatial pattern of surface dust concentration from model
output (shading; right column) and the ratio between modeled and RSMAS station
observed surface dust concentration (color triangles, with upward triangles indicating
overestimation and downward triangles indicating underestimation). 16 stations were
used, and numbers in each triangle (right) and grey dots (left) indicate the stations. The
one-one, one-two and one-five lines are plotted in solid, dashed and dash-dotted lines in
the scatter plots.





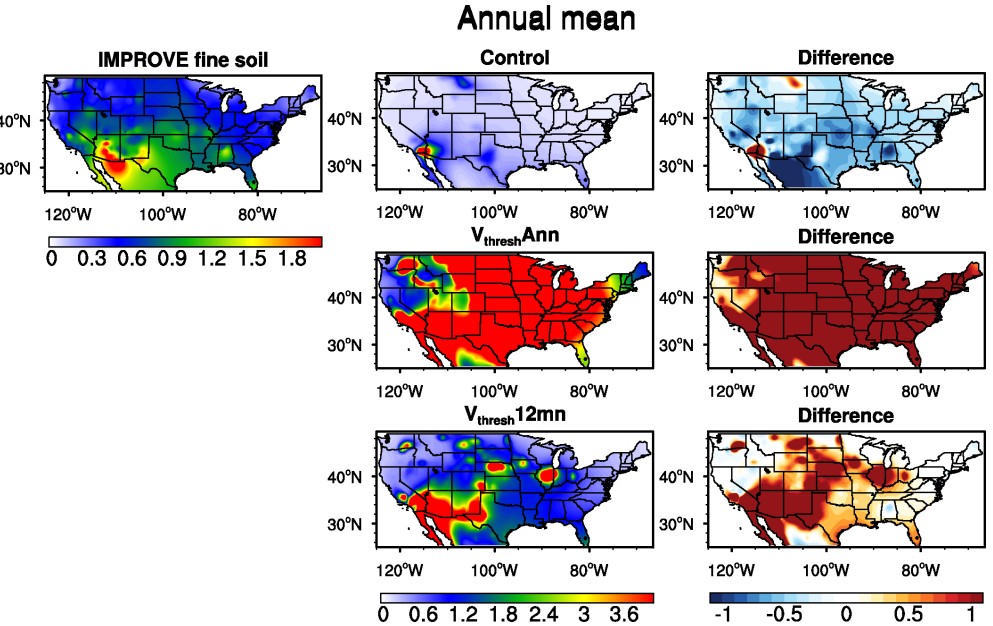

Figure 9. Annual mean surface fine dust concentration ($\mu$g m$^{-3}$) from IMPROVE stations
(left column) and three simulations (middle column) and the differences between model
and observation (right column) for 2002-2015.




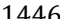

## Dust optical depth (2003-2015)

### (a) Sahel

### (b) Sahara

### (c) Arabian Peninsula

### (d) N. China

### (e) India

### (f) US

### (g) S. Africa

### (h) S. America

### (i) Australia


Figure 10. Seasonal cycle of DOD from MODIS (black), the Control (grey), $V_{thresh}12mn$
(orange), and $V_{thresh}Ann$ (yellow) runs, and gridded AERONET SDA COD (blue)
averaged over nine regions. The annual mean of each dataset in each region is listed on
the top of the plot.




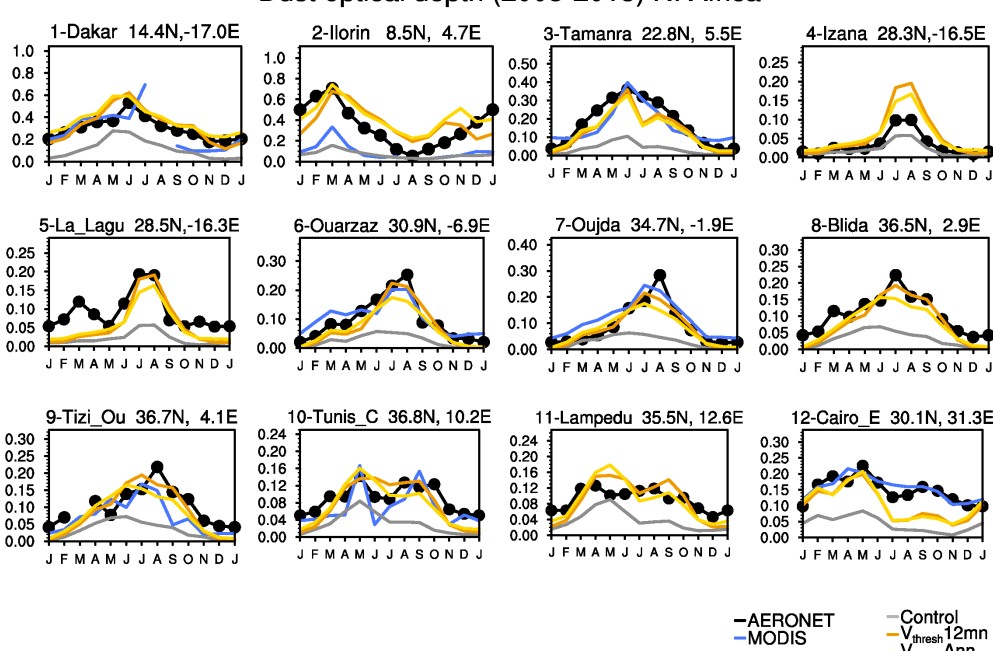


Figure 11. Seasonal cycle of DOD over 12 AERONET SDA sites (see Fig. S5 in the
Supplement for locations) from the Control (grey), $V_{thresh}12mn$ (orange), and $V_{thresh}Ann$
(yellow) simulations, along with DOD from MODIS (blue), and COD from AERONET
(black dotted line). All values are averaged over 2003-2015. The location (lat/long) and
the name (due to space, only first seven characters are shown) of the sites are listed at the
top of each plot.





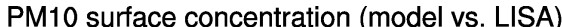

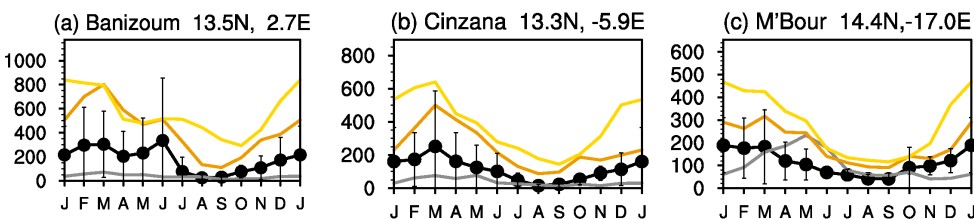

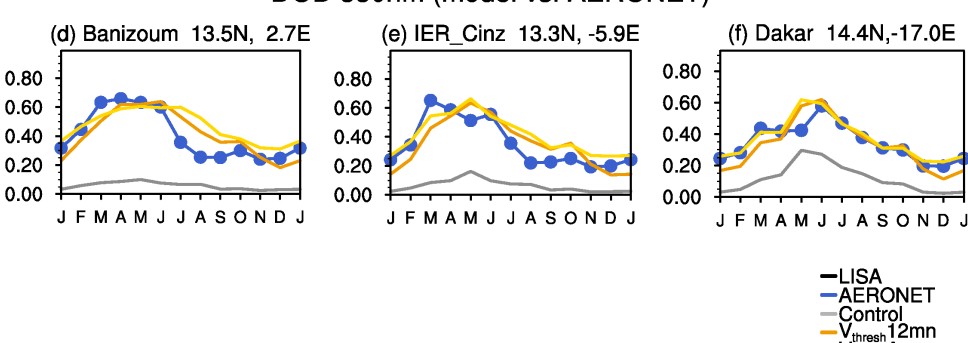

Figure 12. (a)-(c) Seasonal cycle of $PM_{10}$ surface concentration (black) over three sites
from the LISA project, along with $PM_{10}$ surface dust concentration from the Control
(grey), $V_{thresh}12mn$ (orange), and $V_{thresh}Ann$ (yellow) simulations. Error bars are ± one
standard deviations of daily mean in each month averaged over 2006-2014. Unites: µg m⁻
³. (d)-(f) seasonal cycle of DOD (550 nm) from three AERONET sites co-located with
LISA sites (blue) versus that modeled by the Control (grey), $V_{thresh}12mn$ (orange), and
$V_{thresh}Ann$ (yellow) simulations.





## Case Study (Oct.17-19, 2012)

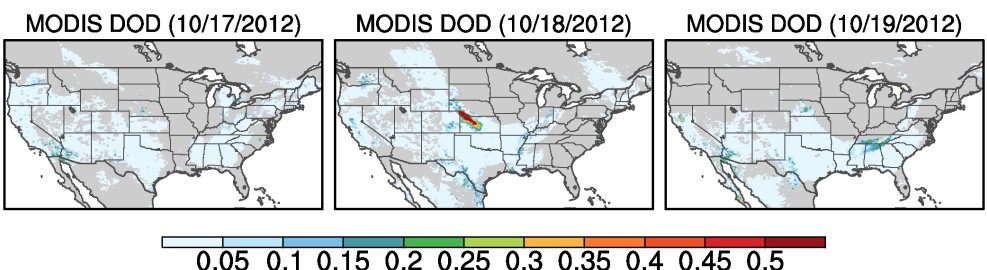

### V$_{thresh}$12mn

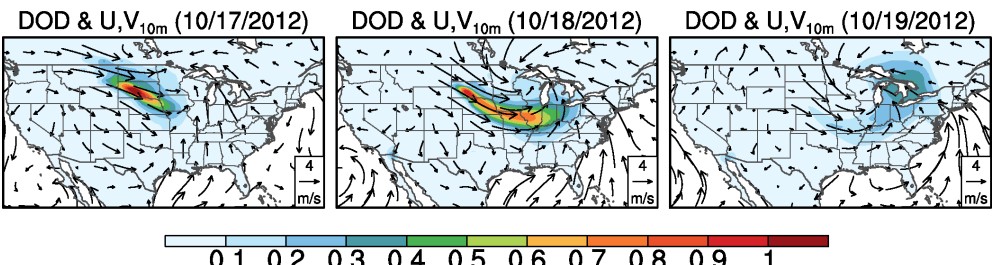

Figure 13. Daily DOD from MODIS (top panel), daily DOD simulated by the V$_{thresh}$12mn
run along with anomalies (with reference to the 2000-2015 mean) of surface wind vectors
(m s$^{-1}$; bottom panel) from Oct. 17$^{th}$ to Oct. 19$^{th}$, 2012. Only DOD over land is shown.
Missing values in MODIS DOD (top panel) are plotted in grey shading.





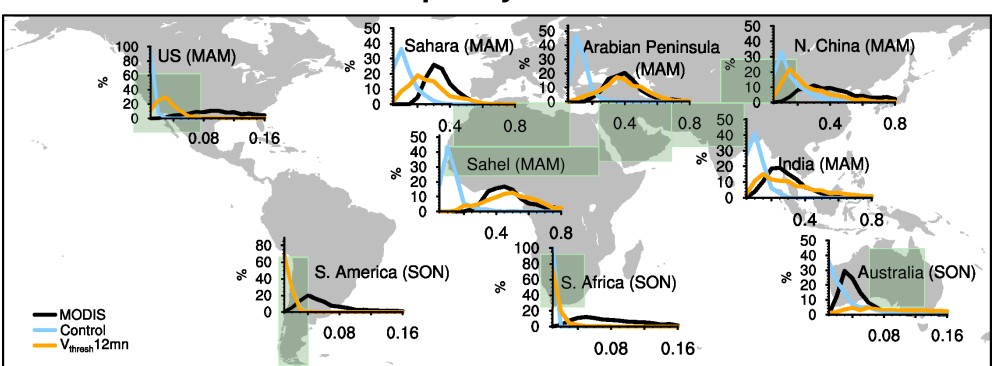

Figure 14. Frequency (%) distribution of regional averaged daily DOD from MODIS
(black) versus that from the Control (light blue) and $V_{thresh}12mn$ (orange) simulations for
the Sahara, the Sahel, the Arabian Peninsula, northern China, India, western to central
U.S., South America, South Africa, and Australia from 2003 to 2015. X-axis denotes the
ranges of DOD, and y-axis is percentage of occurrence. The light green boxes denote the
averaging areas. For regions in the Northern Hemisphere frequency in MAM is shown,
while for regions in the Southern Hemisphere frequency in SON is shown.