# Peer review of "Retrieving the global distribution of threshold of wind erosion from satellite data and implementing it into the GFDL AM4.0/LM4.0 model"

_Atmospheric Chemistry and Physics, 2019_

## Referee Comment (RC1) · Anonymous Referee #1 · 29 Apr 2019

The comment was uploaded in the form of a supplement:
https://www.atmos-chem-phys-discuss.net/acp-2019-223/acp-2019-223-RC1-supplement.pdf

---

## Referee Comment (RC2) · Anonymous Referee #2 · 23 May 2019

This is an interesting paper that produces the first estimation of the global distribution of threshold wind speeds for wind erosion (dust aerosol emission). They do so by combining a calculation of the frequency of dust events per grid box with a probability distribution of wind speeds per grid box from a reanalysis product (NCEP/NCAR). They then implement their estimation of threshold wind speeds into a global model and study the results relative to a control run with a globally-constant threshold wind speed. The paper is overall well-written and easy to follow, and the results could be important because they could help advance dust models beyond the use of a globally constant threshold friction velocity. However, I think there are some important issues with the methodology, the interpretation of the retrieved threshold wind speeds, and with inter-

preting the results from the global model. The paper would need substantial revisions. Comments follow below.

Main comments:

- A major weakness of the methodology is that it equates high dust AOD in a gridbox with the occurrence of dust emission. This causes problems in their methodology because it causes advected dust to be interpreted as emitted dust, and thus results in an underestimation of the dust emission threshold. Since there are large differences in advected dust between regions – for instance areas in major dust regions are bound to be more affected by advected dust – this problem could cause potential biases in the retrieved threshold wind speed. Although the authors commendably acknowledge the problem (e.g., on line 340-2), the magnitude of this bias is not investigated. And unfortunately, without a reasonable analysis of the magnitude of this bias, I do not think the authors can conclude that the threshold wind speed in the Sahel is actually lower than in Northern Africa. And similarly, it is not clear that the lower threshold in the major source regions (e.g., the Sahara) than in the more marginal regions (e.g., the US) is real, or is a result of this bias. In fact, both these results are consistent with the anticipated effect of this bias, as the authors acknowledge for the Sahel. Therefore, the authors need to add an analysis that reasonably bounds the effect of this bias. Perhaps the authors could analyze the wind speed threshold in different regions, conditional on the DOD in the surrounding regions, in order to try to quantify and bound this bias?

- I also think the interpretation of the differences between threshold wind speed must be improved. Of relevance here is that wind speed itself is not the main explanatory variable for dust fluxes. Rather, this is the wind stress on the surface as quantified by the friction velocity, which is linked to the 10m wind speed through the aerodynamic surface roughness. There are strong experimental constraints on the threshold friction velocity above which surface particles become mobile and dust emission starts (e.g., Shao, 2008). It is therefore very relevant what the NCEP/NCAR surface roughness in the different source regions is: do differences in the roughness between source regions

explain the differences in the threshold wind speed? Are threshold wind speed variables substantially correlated with the roughness values used in NCEP/NCAR for each grid box? The authors can also use the surface roughness to determine the distribution of threshold friction velocities for the different regions, which is more fundamental and thus more useful to the community. Another important consideration that follows from this above concern is that, since it's the friction velocity (and wind stress) that drives dust fluxes, the roughness used in GFDL should match the roughness used in the NCEP/NCAR reanalysis. Is this the case?

- Similarly, the authors should investigate differences in other parameters that determine the threshold friction velocity (and 10m wind speed), namely soil moisture, vegetation, and soil texture. If the authors can provide plausible physical reasons for the variations between the threshold wind speed between the regions, that would also help alleviate the concern that their results might be primarily driven by biases arising from using high DOD as a proxy for dust emission (previous comment).

- The rationale for implementing the retrieved threshold wind speed into the GFDL model is not made very clear in the paper, but I assume it is to try and show that using the retrieved threshold wind speed improves GCM simulations of the dust cycle. If so, although the analysis presented is interesting and draws on a commendably wide variety of data, it has some important problems that need to be addressed. First, the proportionality constant in the dust emission equation (Eq. 3) is not constrained by physics (i.e., there's no reason it should be 0.75e-9 ug/s2/m5 instead of 1e-9 or 0.1e-9 ug/s2/m5), and presumably C was set at an earlier stage by maximizing agreement against observational data. Therefore, the fact that using the retrieved threshold wind speeds reduces the underestimation of DOD and dust concentration is not an indication that the retrieved threshold wind speeds actually improve the realism of the model simulation. You would get the same effect simply by increasing the (unconstrained) value of C. The authors should therefore compare apples to apples by tuning the simulations to the same global loading or DOD, and then compare against the AERONET

and other data. This is especially important because using the retrieved threshold wind speeds results in a very large (and again, arbitrary, because C is unconstrained) increase in emissions by a factor of $\sim$4 (Table S2).

- Another problem with the model comparisons against data is that its interpretation requires more rigorous statistics. Keeping in mind the previous comment that the absolute values of DOD and concentration are arbitrary because the emission proportionality constant is unconstrained, the authors would need to show statistically significantly increased correlations between the model and data in order to conclude that the retrieved threshold wind speeds improve the model realism. Otherwise, I do not think the conclusion in the abstract and the paper that the retrieved threshold wind speed improve the simulation can be supported. Correlations are reported in Figs. 4 and 5, and I'm guessing that the improvement is large enough that it's statistically significant, but this ought to be shown. Correlations are not currently reported for the varied results in Figs. 8 – 14, so should be added.

Other comments:

- Line 2: I'd suggest saying "many" instead of "most", as I believe most models at least account for the effect of soil moisture on the threshold wind speed.

- Do you have a sense of how sensitive your results are to the particular reanalysis product used?

- Line 304: it seems hard to imagine that snow cover of 0.2% would prevent or substantially reduce the occurrence of wind erosion. Please provide support for this assumption.

- Line 311-2: "soil moisture ranging from 1.01 to 11.2 kg kg-3"; the units here are incorrect, and I think the number is much too high if the intended unit was kg of water per kg of soil.

- Line 317: I don't think it makes sense to only pick out the daily maximum surface

wind speed when you have wind speeds at 6-hours resolution. You could either argue that the DOD is a product of emission that occurs over a longer time period and thus use winds at all time steps, or you could argue that you are using DOD as a proxy for emissions in the moment and thus use the wind speed closest to your DOD observation (presumably noon since overpasses are at 10:30 am and 1:30 pm). But using the daily maximum does not make sense to me.

- The authors use a threshold DOD of 0.2 over the major source regions of North Africa, the Middle East, etc, which is consistent with previous work in Ginoux et al. (2012). But they use a threshold DOD of only 0.02 in lesser source regions such the US, South America, etc. This is a very large difference of a factor of 10, and seems rather arbitrary. Could the authors either provide an analysis of the sensitivity of their results to this choice or use the actual frequency distribution of DOD in the different source regions to inform these thresholds?

- Section 3.2.3: How are you obtaining AERONET data as a gridded product since data density is so sparse in most dust source regions?

- It's not clear to me whether the control run accounts for the effects of soil moisture on the threshold wind speed or whether it truly uses a constant threshold wind speed, regardless even of soil moisture content. Could you clarify?

Editorial comments:

- Line 57: Since wind speeds are a function of height, please note what these wind speeds refer to.

- Since the methodology is quite involved and lengthy, I recommend you provide an overview of your methodology in a paragraph at the beginning of section 2 to make the paper easier to read.

- 182-184: Please provide more info or a citation to a peer-reviewed paper here for the reader to understand how LAI is calculated.

- Line 254-5: This is a common assumption in using the dust concentration data, so you could support this by citing precedent in previous studies.

- Section 3.3: I think this section would be placed more logically before the case study.

- Figure 8: since the data here span 3 orders of magnitude, providing statistics in linear space is not very meaningful as it weighed heavily toward the large concentration data. Please provide statistics in logarithmic space.

- Fig. 14: What is the bin spacing on the horizontal axis? The reader needs that to interpret the percentage given on the vertical axis.

---

## Author Comment (AC1) · 28 Oct 2019

Department of Geography and Atmospheric Science
University of Kansas
Jayhawk Boulevard
Lawrence, KS 66045
Email: bpu@ku.edu

Dr. Yves Balkanski
Institut Pascal et
IPSL/LSCE (Laboratoire des Sciences du Climat et de l'Environnement)
CEA-CNRS-UVSQ-UPSaclay UMR 8212
L'Orme des Merisiers - Bat 714, pce 1012
91191 Gif sur Yvette Cedex, FRANCE

October 27th, 2019

Dear Editor Balkanski,

We have submitted a revised paper entitled "Retrieving the global distribution of threshold of wind erosion from satellite data and implementing it into the GFDL AM4.0/LM4.0 model" by B. Pu, P. Ginoux and co-authors for consideration for *Atmospheric Chemistry and Physics*. The helpful comments from two anonymous reviewers are sincerely appreciated. Our replies to each reviewer's comments are attached. We also made some edits in the manuscript.

We gratefully appreciate your time and consideration!

Sincerely,

Bing Pu

Review of the ACPD manuscript "Retrieving the global distribution of threshold of wind erosion from satellite data and implementing it into the GFDL AM4.0/LM4.0 model" by Pu et al.

We thank the reviewer for very helpful comments. We reply to your comment (in Italic) below.

*The article by Pu et al. describes a new data set for the threshold wind velocity for dust emission and shows the impact on dust aerosol simulated with the GFDL model. The authors used a comprehensive collection of observational data to approach the problem. In principle, the contribution is relevant to the field, since modeling dust aerosol is fraught by uncertainty. I have, however, concerns that should be address prior to publication of the article. These are the unclear description of the method, the lack of an uncertainty assessment for the retrieval, as well as the need for a comparison to independent data and citing of relevant literature. In the following, I provide more details.*

In addition to reply each of the following comments, we also edited the manuscript to better address your comments and suggestions.

*Main comments:*
*1) The description and uncertainties of the method are unclear. The article suffers from an unclear description and partly missing information on the retrieval technique. Moreover, the value of the article would be substantially improved when the uncertainty in the retrieval would be quantitatively assessed. The many threshold criteria in the retrieval currently cast some doubt on the robustness of the retrieval when these values would be slightly changed.*

We modified lines 335-393 to improve the clarity of the retrieval method and added section 2.3 and Tables 2-3 to discuss and quantify the uncertainties associated with slight changes of retrieval criteria and selection of surface wind datasets. We found small changes in soil moisture, LAI, and snow coverage do not change the derived $V_{threshold}$ much, within 1 m s$^{-1}$ over most regions. The results are more sensitive to $DOD_{thresh}$ and the selection of surface winds from reanalysis products (Tables 2-3). The uncertainty of DOD frequency distribution and $V_{threshold}$ associated with transported dust is also discussed over North Africa (lines 414-426, 432-438). Global $V_{threshold}$ using $DOD_{thresh}$ =0.2 (or 0.02) and 0.5 (or 0.05) are further compared and discussed in section 3.1.

*2) The article needs more comparisons to existing works. The current article does not acknowledge other existing treatments of the threshold of wind erosion for global models. For instance, Cheng et al. (2008), Jones et al. (2011) and Rieger et al. (2017) do not prescribe globally constant threshold wind speeds for dust emission, but parameterize it with dependencies on other variables. These are the global models ECHAM-HAM, HadGEM2-ES, and ICON-ART. Such studies should be cited and used for comparison of the new development in the GFDL model.*

Thanks a lot for your suggestions. We added lines 64-68 to better address this question: "On the other hand, some models, such as the ECHAM-HAM, HadGEM2-ES, and ICON-ART, parameterize the constant dry threshold friction velocity (usually a function of soil particle size, soil and air density) or threshold wind velocity with dependencies on soil moisture, surface roughness length, and vegetation coverage (e.g., Takemura et al. 2000; Ginoux et al. 2001; Zender et al. 2003; Cheng et al. 2008; Jones et al. 2011; Rieger et al. 2017)."

      While Cheng et al. (2008), Jones et al. (2011) and Rieger et al. (2017) all parameterize the threshold friction velocity in different models with dependencies on other variables, such as soil moisture, surface roughness length, and vegetation coverage, the dry friction velocities used in the models are largely based on constant values such as air and soil density and soil particle size (e.g., Eq. 3 of Rieger et al. 2017; Eq. 1 of Cheng et al. 2008; Eq. 3 of Woodward 2011).

*Specific comments:*
*P1. L.37: "enhancing net radiant energy loading" Use a physically better phrase.*
      We changed "enhancing net radiant energy loading" to "enhancing net radiation".

*P1. L46: "the life cycle of dust" -> "the life cycle of dust aerosols"*
      Done.

*P.6 L124-126: "We require that the single scattering albedo at 470 nm to be less than 1 for dust due to its absorption of solar radiation. This separates dust from scattering aerosols, such as sea salt." The single scattering albedo is by definition smaller than 1. So it will not separate dust and sea-salt aerosol. This statement leaves me puzzled about the adopted method for obtaining dust aerosol optical depth from MODIS. The method needs to be revised and the description clarified. The remaining sentences of the paragraph give more details, but it is not obvious how the method works without reading all the other publications. My recommendation is giving a more concrete and easier to follow description of the method here. For instance, how is dust separated from other aerosols and how are dust sources identified. Also provide important numbers, e.g., for the separation of fine-mode vs. dust aerosols and the definition of high-resolution.*

      Lines 136-146 are modified to better address the comment. We used single-scattering albedo at 470 nm to be less than 0.99 for dust, as the single-scattering albedo of sea salt is close to 1. The resolution of the MODIS products is 0.1° by 0.1°. The retrieval method is summarized in Eq. 1.

*P.6 L.134-137: What does a flag of QA=1 and QA=3 imply for the quality of the data?*
      We modified lines 151-153 to clarify this. For MODIS Deep Blue AOD products, quality assurance flag (QA) equals 0, 1, 2, or 3 (Hsu et al. 2013). QA=0 indicates no retrieval, while QA=1 indicates lowest quality of retrieved AOD, and QA=3 implies the highest quality.

*P.6 L.139-143: I understand combining the morning and afternoon measurements is the best we can do, but the text should acknowledge that the location and amount of dust emission typically changes between the morning and afternoon. Peak contributions from convective storms would be missed due to the temporal resolution. A relatively large number of literature assesses the diurnal cycle of dust emission and some of those studies could be cited here. My point is that the strengths and weaknesses of the method need to*

*be named as far as it is currently known. This also applies to the other satellite products (soil moisture, snow cover, LAI) introduced in the next paragraphs.*

We added lines 162-165 to better address this issue: "Note that due to the temporal coverage of MODIS products, the diurnal variations in dust (e.g., Orgill and Sehmel 1976; Mbourou et al. 1997; Knippertz et al. 2008; Schepanski et al. 2009) are not included in current study." Later in lines 772-773, we mentioned: "Diurnal variability of dust emission and short-duration events such as haboobs are also not included. " Lines 184-185, 207-208 are also added to discuss the uncertainties associated with soil moisture and LAI products.

*P.8 L.177: "Vegetation can protect soil (...)" -> Vegetation protects soils (...)*
Done.

*P.8 L. 182-184: The description of the data set is not published. At least a short description of the retrieval is needed and also a statement on where one can access or request that data.*

We modified lines 204-205 to address this point. Details about LAI retrieval can be found from Yan et al. 2016a, b. We mentioned that the data was obtained via personal communication with Ranga Myneni and Taejin Park in Boston University in 2016 in text. In the Acknowledgement we added "MODIS LAI data may be requested by contacting Dr. Ranga Myneni at Boston University".

*P. 9 L. 186-187: A six hourly resolution of the winds does not sufficiently resolve their diurnal cycle and hence their effect on dust emission. Again, the diurnal cycle of dust emission is an issue here, but for the model data we could fix it.*

We mentioned in lines 162-165 and 772-773 that diurnal cycles are not included in the analysis. 6-hourly winds from the NCEP are selected because surface winds in the model are nudged toward NCEP winds, and we would like to use a reanalysis that is close to the climatology of the model. Similar methods can be applied to other reanalyses with higher temporal resolutions, e.g., hourly surface winds from the ERA5 as discussed in section 2.3. On the other hand, whether model can faithfully capture the diurnal cycle of dust also depends on the dust emission scheme and model's capability to simulate high-speed winds and mesoscale convective system, which are beyond the scope of this study.

*Section 2.1.2: Why did you choose two different re-analyses? Did you also consider using MERRA?*

As we mentioned above, the NCEP reanalysis is chosen because surface winds are nudged toward it. For soil temperature at the first layer, we use the ERA-Interim, which has higher spatial resolution. The horizontal resolution of ERA-Interim is about 0.7° and is comparable with that of MERRA (Rienecker et al. 2011) or MERRA-Land (Reichle 2012) on a 1/2° by 2/3° grid. While MERRA surface temperature is found have a relatively large bias (>3°C) in comparison with AMSR-E temperature in desert region (Yi et al. 2011), MERRA-Land surface soil moisture is found to have slightly lower skill than the ERA-Interim when comparing with SCAN in situ surface moisture (Reichle et al. 2011). Later, we also used surface winds from the ERA-Interim and ERA5 to examine the sensitivity of $V_{threshold}$ to the selection of reanalysis products.

*P.9 L. 192: "closet" -> closest*
    Done.

*P.9 L. 205: "coarse mode AOD" What is the radius for separating coarse and fine-mode AOD in your work?*
    Based on O'Neill et al. (2003) coarse-mode AOD has a radius greater than 0.6 μm. We added this to line 240.

*P.10 L. 209-210: Three years is a very short time period for a climatology, especially in light of the strong year-to-year variability in dust aerosol burden. I agree that as little data as possible should be removed. However, I recommend giving an estimate of the uncertainty, e.g., try a stricter criterion and compare the climatologies.*
    Thanks for you suggestion. We found a stricter criterion will result a smaller sample size and the results won't change much. For instance, if a minimum record length of five years is used, there will be 225 sites for SDA COD and 263 sites for AOT (instead of 313 and 351 stations as shown in Fig. 5). If seven-year is used as a requirement, there will be 156 SDA COD sites and 195 AOT sites. The climatologies of using five or seven years records as a criterion are shown in Figures R1 and R2, respectively. Results are very similar the climatology using three years as a criterion (Fig. 4).

[Figure]

Figure R1. Same as Fig. 4 but using stations with at least five years of records.

[Figure]

Figure R2. Same as Fig. 4 but using stations with at least seven years of records.

*P.10 L.227: Refer to the section of the article.*

We removed lines 259-262 in the revision (lines 224-227 in the previous version of the manuscript).

*Section 2.1.3: Consider showing a map with the location of the different stations used for this research. You could use color to indicate the record length of the stations.*

We added the length of records of the RSMAS stations to Table S1 in the Supplement, where the latitude/longitude of each station is also shown. The record length of AERONET AOT and SDA data are now added to Figure S6 in the Supplement. The location and length of records from the IMPROVE can be found from Pu and Ginoux (2018; Fig. S1), while the location of three LISA sites can be found in Fig. S7 in the Supplement of this paper. All available hourly LISA station data from 2006 to 2014 are used to calculate daily mean and then monthly mean as mentioned in lines 324-325.

*P.12 L. 254-255: " (…) assume that the climatology of the surface dust concentrations do not change greatly from the 1980s to the 2000s" Why is this a reasonable assumption?*

I agree this is not necessarily a good assumption. Lines 288-292 are modified to better address this point: "Note that since most station records end earlier than 1998, the dataset largely represents the climatology during the 1980s and 1990s. Thus the discrepancies between model output and the RSMAS data include both model biases and the difference in surface dust concentration from the 1980s to the 2000s." Despite the uncertainties, the RSMAS dataset has been widely used for model validation. For instance, Huneeus et al. (2011) used the climatology of the data to validate AeroCom model simulations in 2000.

*P.14 L.303-307: Why did you choose these thresholds? For instance, why not a snow cover of 0% and an LAI of 0? I can imagine this is due to fractional difference within a grid box, but it is unclear whether a slight change in the thresholds would have a big effect on the results. Maybe you could test it for obtaining more confidence in the results.*

     As mentioned in lines 347-353, similar criteria have been used to detect or confine dust source regions by different studies. For instance, "LAI less than 0.3 has been used as a threshold for dust emission in the Community Land Model (Mahowald et al., 2010; Kok et al., 2014a)". We also added sensitivity tests in section 2.3 better quantify how the small variations in the retrieval criteria may affect the retrieved $V_{threshold}$.

*P.15 L.321-333: I understand that you choose different background dust AODs per region, but where does 0.2 and 0.02 come from? Could you use the minimum in dust AOD from daily values in your MODIS climatology to accurately compute the background values?*

     MODIS DOD has small values near zero (see Fig. S1 form Pu and Ginoux 2017), so it is difficulty to use the minimum value in DOD to compute background aerosol values. $DOD_{thresh}$=0.2 was used by Ginoux et al. (2012) to distinguish dust events from background aerosols. We used $DOD_{thresh}$=0.02 for less dusty regions, such as North America, South Africa, South America, and Australia, largely because dust emission in these regions are at least ten times smaller than that from dusty regions such as North Africa (Huneeus et al. 2011). As shown in Fig. 15, in these less dusty regions, the averaged frequency distribution of DOD peak over much smaller values than dusty regions (lines 748-751). While the selection of $DOD_{thresh}$=0.2 (or 0.02) is empirical, we also tested $DOD_{thresh}$=0.5 (or 0.05), and results are discussed in section 3.1.

*P.15 L. 339-343: I appreciate the general acknowledgement of potential uncertainty in the thresholds. I think a quantitative assessment of the uncertainty would substantially strengthen your work. You could easily do so by varying the threshold criteria within bounds you perceive reasonable (justified by physical arguments) and show the associated changes in your results.*

     Thanks for your advice. We added section 2.3, Tables 2-3 and modified later discussion in section 3.1 (lines 505-569) to better quantify the uncertainties associated with the varying threshold retrieval criteria.

*P.16 L.365: How was the scaling factor determined?*

     The scaling factor $C$ in the standard version of the AM4.0/LM4.0 was determined by matching the modeled surface dust concentrations with the RSMAS station data. We did not change it in the simulations in order to compare the differences associated with different $V_{threshold}$.

*P.18 L.399: "differences in simulated dynamic vegetation by LM4.0 among the three simulations are actually very small and can be ignored" add that this is the case because of the short simulation when the land use does not change as much as over longer time periods.*
       Done.

*P.18 L.412: What primarily controls the threshold differences between North Africa and Eurasia? A threshold of 3 m s$^{-1}$ is very low and needs an explanation.*
       These lines are removed and Tables 2-3 are added to better quantify the regional difference of $V_{threshold}$. The magnitude of threshold wind erosion is determined by matching the cumulative frequency of DOD at certain $DOD_{thresh}$ level with the frequency distribution of surface wind speed. Therefore, regions with higher DOD frequency (e.g., high FoO in Figs. 1a-e) generally have lower threshold of wind erosion. As discussed in sections 2.3 and 3.1, value of $V_{threshold}$ in North Africa is lower in comparison with previous station based estimations, and this is largely associated with lower surface wind speed in the NCEP1 reanalysis and the ignorance of the contribution of transported dust to total DOD. Increasing $DOD_{thresh}$ to 0.5 can increase annual mean $V_{threshold}$ over North Africa to 4.9 ~7.6 m s$^{-1}$ (Table 2). However, despite the relatively low value of $V_{threshold}$ in North Africa, we found the spatial and temporal varying $V_{threshold}$ largely improve the simulation of DOD spatial pattern and seasonal cycle over North Africa in the AM4.0/LM4.0 model.

*P.19 L.435: "weed" -> wind*
       Done.

*P.19 L.423- 439: A discussion is useful, but the results keep me thinking of the potential impact of the threshold choices in the retrieval. This is not picked up in the discussion of your lower threshold velocities than in previous studies.*
       We added section 2.3 and Tables 2-3 to quantify the uncertainties associated with slight changes of retrieval criteria, and modified lines 538-540, 544-545, 562-569 in section 3.1 to discuss these uncertainties when comparing with previous studies.

*P.25 L.572: Harmattan winds are important in winter and spring. Fiedler et al. (2015) provide a complete climatology of dust aerosol associated with the Harmattan.*
       We modified line 697: "...are associated with the dry northerly Harmattan wind in boreal winter and spring..." and added the citation of Findler et al. (2015).

*P.27 L. 608: "storm centers a bit" -> storm center is located*
       Done.

*Section 3.3: It would be useful to compare against independent data sets already published since both the model and the observational estimates have been newly developed in the current article. Relevant works are for instance Schepanski et al. (2007) and Evan et al. (2015).*
       Since section 3.3 and Fig. 15 only show regional averaged frequency distribution of DOD instead of spatial pattern, we add discussion in lines 387-393 to compare the FoO in Fig. 1 with dust emission frequency over North Africa from Evans et al. (2015) and frequencies of dust source activation from Schepanski et al. (2007).

*Figure 8: Refine the color scale for the surface concentration in the dust belt. The same red shading does not allow a comparison of the results in the dust regions.*
  Done.

*Figure 10: Except for India, US and South America, the difference in the annual cycles in Vthres12mn and VthresAnn is very small. It suggests that the month-to-month variation in threshold wind velocities does not have a large impact on the climatological mean dust aerosol optical depth in main dust sources. Is this primarily so because the variations in soil moisture of deserts are small or what explains the similarity?*
  We agree that in Fig. 10 (now Fig. 11) expect India, U.S. and South America the differences between $V_{thresh}Ann$ and $V_{thresh}12mn$ are small. The small season variations in soil moisture in dust source regions (largely arid or semi-arid regions) may play a role. On the other hand, the differences between $V_{thresh}Ann$ and $V_{thresh}12mn$ simulations are larger when comparing surface dust concentration (Figs. 9, 10,13a-c).

*Figure 14: Add VthresAnn.*
  We show results form $V_{thresh}Ann$ here in Figure R3. The regional mean DOD frequency distribution from the $V_{thresh}Ann$ simulation (yellow) is largely similar to that from the $V_{thresh}12mn$ (orange), except over the U.S., India, and South America, where DOD peaks at higher values, i.e., slightly closer to the peaks in MODIS. This is consistent with higher DOD in these regions (Fig. 11) in comparison with the $V_{thresh}12mn$ simulation. Since results from $V_{thresh}12mn$ generally show better agreement with station observations and MODIS DOD (e.g., Figs. 8-13), we chose to focus on the results form the $V_{thresh}12mn$ in section 3.2.4 and 3.3 (Figs. 14-15).

**Frequency of DOD**

[Figure]

Figure R3. Same as Fig. 15 but also include the DOD distribution frequency from the $V_{thresh}Ann$ simulation (yellow line).

We thank the reviewer for very helpful comments. We reply to your comment (in Italic) below.

*This is an interesting paper that produces the first estimation of the global distribution of threshold wind speeds for wind erosion (dust aerosol emission). They do so by combining a calculation of the frequency of dust events per grid box with a probability distribution of wind speeds per grid box from a reanalysis product (NCEP/NCAR). They then implement their estimation of threshold wind speeds into a global model and study the results relative to a control run with a globally-constant threshold wind speed. The paper is overall well-written and easy to follow, and the results could be important because they could help advance dust models beyond the use of a globally constant threshold friction velocity. However, I think there are some important issues with the methodology, the interpretation of the retrieved threshold wind speeds, and with interpreting the results from the global model. The paper would need substantial revisions. Comments follow below.*

In addition to reply each of the following comments, we also edited the manuscript to better address your comments and suggestions.

*Main comments:*
*- A major weakness of the methodology is that it equates high dust AOD in a gridbox with the occurrence of dust emission. This causes problems in their methodology because it causes advected dust to be interpreted as emitted dust, and thus results in an underestimation of the dust emission threshold. Since there are large differences in advected dust between regions – for instance areas in major dust regions are bound to be more affected by advected dust – this problem could cause potential biases in the retrieved threshold wind speed. Although the authors commendably acknowledge the problem (e.g., on line 340-2), the magnitude of this bias is not investigated. And unfortunately, without a reasonable analysis of the magnitude of this bias, I do not think the authors can conclude that the threshold wind speed in the Sahel is actually lower than in Northern Africa. And similarly, it is not clear that the lower threshold in the major source regions (e.g., the Sahara) than in the more marginal regions (e.g., the US) is real, or is a result of this bias. In fact, both these results are consistent with the anticipated effect of this bias, as the authors acknowledge for the Sahel. Therefore, the authors need to add an analysis that reasonably bounds the effect of this bias. Perhaps the authors could analyze the wind speed threshold in different regions, conditional on the DOD in the surrounding regions, in order to try to quantify and bound this bias?*

Thanks for your suggestion. We roughly estimated the influence of transported dust on wind erosion threshold ($V_{threshold}$) in North Africa using a surface DOD (sDOD) data retrieved by combining lidar vertical profiles from CALIOP and MODIS Dust Optical Depth in section 2.3 (lines 414-426). As shown in Table 2, $V_{threshold}$ over the Sahel (6.05 vs. 3.21 m s$^{-1}$) and Sahara (7.66 vs. 4.61 m s$^{-1}$) from sDOD are higher than that from DOD directly. Here $V_{threshold}$ in the Sahara is still higher than that in the Sahel.

This is consistent with the findings of Chomette and Legrand (1999) and Cowie et al. (2014), who also showed wind erosion threshold was higher in most part of the Sahara than the Sahel.

 We also quantify and discuss the uncertainties of $V_{threshold}$ associated with slight variations in retrieval criteria including levels of soil moisture, LAI, snow coverage, $DOD_{thresh}$, and surface wind speed from different reanalyses in section 2.3 and added Tables 2-3 to better display the regional difference of retrieved $V_{threshold}$. In most case, we notice the $V_{threshold}$ in the Sahara is lower than in the U.S., except using sDOD in North Africa and when we used $DOD_{thresh}$ =0.5 (and 0.05 for less dusty regions).

*- I also think the interpretation of the differences between threshold wind speed must be improved. Of relevance here is that wind speed itself is not the main explanatory variable for dust fluxes. Rather, this is the wind stress on the surface as quantified by the friction velocity, which is linked to the 10m wind speed through the aerodynamic surface roughness. There are strong experimental constraints on the threshold friction velocity above which surface particles become mobile and dust emission starts (e.g., Shao, 2008). It is therefore very relevant what the NCEP/NCAR surface roughness in the different source regions is: do differences in the roughness between source regions explain the differences in the threshold wind speed? Are threshold wind speed variables substantially correlated with the roughness values used in NCEP/NCAR for each grid box? The authors can also use the surface roughness to determine the distribution of threshold friction velocities for the different regions, which is more fundamental and thus more useful to the community. Another important consideration that follows from this above concern is that, since it's the friction velocity (and wind stress) that drives dust fluxes, the roughness used in GFDL should match the roughness used in the NCEP/NCAR reanalysis. Is this the case?*

 The reviewer is wondering if the differences in surface roughness in the NCEP/NCAR reanalysis can explain the differences in the threshold wind speed between source regions. The distribution of threshold wind is determined by matching the frequency distribution of DOD at certain level of $DOD_{thresh}$ with the frequency distribution of surface wind speed from the NCEP/NCAR reanalysis. The roughness length ($z_0$) in the NCEP/NCAR reanalysis came from the Simple Biosphere Model (Kalnay et al. 1996). It is calculated based on height of the top and base of the canopy, the height of the maximum leaf area density, leaf drag coefficients, time-varying leaf area index, and ground roughness length for each vegetation table (Table 3 and Fig. 7 from Dorman and Sellers, 1989). The spatial pattern shows dependence on vegetation type (Fig. 7 of Dorman and Sellers, 1989), and has little variation over bare ground. While roughness length plays an important role in the calculation of surface momentum transfer and friction velocity, it does not directly related to the spatial pattern of the $V_{threshold}$ we derived.

 The reviewer also suggested calculating the distribution of threshold friction velocity ($u_t^*$) based on surface roughness. While friction velocity has been used in a lot of dust emission schemes and can be approximated with surface roughness, developing a global distribution of $u_t^*$ and compare with available observations is beyond the scope of this study. Also, instead of using $V_{threshold}$ and surface roughness, it is probably better to use the frequency distribution of $u^*$ and DOD to derive the $u_t^*$, i.e., using a similar method as proposed here.

The dust emission scheme (Ginoux et al. 2001) in the GFDL AM4.0/LM4.0 uses surface wind speed, rather than friction velocity. We did not tune surface roughness in the GFDL model toward that in the NCEP/NCAR reanalysis, which is calculated by the turbulent transfer model in the Simple Biosphere Model (Dorman and Sellers, 1989; Sellers et al. 1989). However, in our simulations, surface wind speeds are nudged toward the surface wind of the NCEP/NCAR reanalysis with a relaxation timescale of 6 hours.

*- Similarly, the authors should investigate differences in other parameters that determine the threshold friction velocity (and 10m wind speed), namely soil moisture, vegetation, and soil texture. If the authors can provide plausible physical reasons for the variations between the threshold wind speed between the regions, that would also help alleviate the concern that their results might be primarily driven by biases arising from using high DOD as a proxy for dust emission (previous comment).*

We added section 2.3 and Tables 2-3 to better discuss the sensitivity of $V_{threshold}$ to retrieval method, reanalysis products, and also the possible biases of using DOD frequency distribution to approximate dust emission in North Africa. We found small changes in soil moisture, LAI, and snow coverage do not change the derived $V_{threshold}$ much, within 1 m s$^{-1}$ over most regions. The results are more sensitive to $DOD_{thresh}$ and the selection of surface winds from reanalysis products (Tables 2-3). The uncertainty of DOD frequency distribution and $V_{threshold}$ associated with transported dust is also discussed over North Africa (lines 414-426, 432-438). Global $V_{threshold}$ using $DOD_{thresh}$ =0.2 (or 0.02) and 0.5 (or 0.05) are further compared and discussed in section 3.1.

Regional differences of $V_{threshold}$ are also better quantified in the Tables 2-3. The spatial and temporal differences of the threshold of wind erosion ($V_{threshold}$) are largely determined by frequency distribution of DOD and surface wind speeds. Therefore, for areas with high dust frequency of occurrence (FoO), e.g., North Africa and the eastern Arabian Peninsula, $V_{threshold}$ is generally lower (Figs. 1e and j). We added discussion in lines 387-393 and modified lines 505-513 to better address this.

Although the overall magnitude of retrieved $V_{threshod}$ using surface winds from the NCEP1 reanalysis is lower than previous station based studies over North Africa. We found the spatial pattern of $V_{threshold}$ —with lower values over the Sahel and slightly higher values over the Sahara —are consistent with results from Chomette and Legrand (1999) and Cowie et al. (2014). The magnitude of retrieved $V_{threshold}$ over northern China is largely consistent with previous studies (Kurosaki and Mikami 2007; Ginoux and Beroubaix 2017).

*- The rationale for implementing the retrieved threshold wind speed into the GFDL model is not made very clear in the paper, but I assume it is to try and show that using the retrieved threshold wind speed improves GCM simulations of the dust cycle. If so, although the analysis presented is interesting and draws on a commendably wide variety of data, it has some important problems that need to be addressed. First, the proportionality constant in the dust emission equation (Eq. 3) is not constrained by physics (i.e., there's no reason it should be 0.75e-9 ug/s2/m5 instead of 1e-9 or 0.1e-9 ug/s2/m5), and presumably C was set at an earlier stage by maximizing agreement*

*against observational data. Therefore, the fact that using the retrieved threshold wind speeds reduces the underestimation of DOD and dust concentration is not an indication that the retrieved threshold wind speeds actually improve the realism of the model simulation. You would get the same effect simply by increasing the (unconstrained) value of C. The authors should therefore compare apples to apples by tuning the simulations to the same global loading or DOD, and then compare against the AERONET and other data. This is especially important because using the retrieved threshold wind speeds results in a very large (and again, arbitrary, because C is unconstrained) increase in emissions by a factor of 4 (Table S2).*

We added lines 441-442, 570-574 to better explain the purpose of implementing the retrieved threshold wind speed into the GFDL model.

The reviewer found $C$ in Eq. 4 is not constrained by physics. Here $C$ is a global tuning factor to adjust the magnitude of dust emission. In the model, surface winds are modulated by the model resolution as well as the model physics parameterizations, which make necessary to use a global tuning factor, assuming that the biases are constant globally. In the default version of the AM4.0/LM4.0, $C = 0.75 \times 10^{-9}$ is obtained by matching modeled dust surface concentrations with RSMAS station records.

We agree with the reviewer that increasing $C$ will increase the magnitude of dust emission and also DOD, which can reduce the bias of underestimation in the Control run. However, as mentioned in lines 476-480: "Here we choose not to retune the dust emission scheme but instead test the usage of $V_{threshold}$, which theoretically provides a more physics-based way to improve dust simulation. We also choose to keep the tuning factor $C$ (Eq. 4) the same in all simulations to better examine the effects of implementing the newly developed $V_{threshold}$. " While tuning $C$ can increase overall dust emission and DOD magnitude, it cannot improve the spatial pattern and seasonal cycle of DOD or surface dust concentrations. We added lines 812-817 to better clarify this point: "The major benefit of using the spatial and temporal varying $V_{threshold}$ is that it improves the simulation of DOD spatial pattern (Figs. 6-7), seasonal cycle (Figs. 11-13), and frequency distribution (Fig. 15) as well as the spatial pattern of surface dust concentrations (Figs. 9-10), which cannot be achieved by simply modifying the global tuning factor ($C$ in Eq. 4) to fit the observations such as surface concentrations or optical depth."

We also conducted a test run to increase dust emission in the Control run (namely, Control II) to about 1232 Tg yr$^{-1}$, which is close the to a previous estimation based on MODIS DOD (1223 Tg yr$^{-1}$; Ginoux et al. 2012) or the AeroCom multi-model median (1123 Tg yr$^{-1;}$ Huneeus et al. 2011). We found the magnitude of DOD slightly increases, e.g., over the Sahel annual mean increases from 0.07 to 0.09, however, there's no improvement in terms of seasonal cycle or spatial pattern, as expected (see discussion in lines 818-825).

We also follow the comments of the reviewer to conduct two other simulations using this enlarged $C$ and 12-month and annual mean $V_{threshold}$ (using $DOD_{thresh}$= 0.5 or 0.05), i.e., V$_{thresh}$12mn II and V$_{thresh}$Ann II simulations, to compare with Control II (see lines 826-837 for details). We choose to use the same $C$ instead of tuning all the simulations to a similar magnitude of global dust emission or DOD. This will help us better attribute the differences among simulations, and also help us quantify the modification on global dust emission/DOD due to the implementation of the $V_{threshold}$. We found similar improvement in DOD seasonal cycle and weaker improvement in DOD

spatial pattern and frequency distribution and surface dust concentrations in $V_{thresh}12mn$ II and $V_{thresh}Ann$ II simulations. This is largely because higher $V_{threshold}$ results in lower global dust emissions in the $V_{thresh}Ann$ II (1961 Tg yr$^{-1}$) and $V_{thresh}12mn$ II simulations (1705 Tg yr$^{-1}$) and overall lower DOD globally. Over Mediterranean coast, Europe, and northern Asia, DOD spatial pattern is not as well captured in the $V_{thresh}12mn$ II run as in the $V_{thresh}12mn$ run, likely due to relatively high $V_{threshold}$ in these regions.

*- Another problem with the model comparisons against data is that its interpretation requires more rigorous statistics. Keeping in mind the previous comment that the absolute values of DOD and concentration are arbitrary because the emission proportionality constant is unconstrained, the authors would need to show statistically significantly increased correlations between the model and data in order to conclude that the retrieved threshold wind speeds improve the model realism. Otherwise, I do not think the conclusion in the abstract and the paper that the retrieved threshold wind speed improve the simulation can be supported. Correlations are reported in Figs. 4 and 5, and I'm guessing that the improvement is large enough that it's statistically significant, but this ought to be shown. Correlations are not currently reported for the varied results in Figs. 8 – 14, so should be added.*

As we mentioned above, the default $C$ in the model is not an "arbitrary" value. It is obtained by matching modeled dust surface concentrations with RSMAS station records.

Following the comments from the reviewer, we show in Table R1 here to demonstration whether the correlations between the $V_{thresh}12mn$ simulations and observational data in comparison with correlations between the Control and observational data are significantly different (or increased) for Figs. 5, 6, 9, 10.

Table R1 Correlations between model output and observational datasets for Fig. 5, 6, 9, and 10.

| Correlations | Figure # | Correlation coefficient (r) | 95% confidence intervals | Significantly different? |
|---|---|---|---|---|
| Control COD vs. AERONET COD | Fig. 5 | 0.68 | 0.62~ 0.74 | Y |
| $V_{thresh}12mn$ vs. AERONET COD | Fig. 6 | 0.84 | 0.80~ 0.87 | |
| Control surface dust vs. RSMAS | Fig. 9 | 0.76 | 0.42~ 0.91 | N |
| $V_{thresh}12mn$ vs. RSMAS | Fig. 9 | 0.72 | 0.35~ 0.90 | |
| Pattern correlation of Control vs. IMPROVE | Fig. 10 | 0.41 | 0.38~ 0.43 | Y |
| Pattern correlation of $V_{thresh}12mn$ vs. IMPROVE | Fig. 10 | 0.55 | 0.53~ 0.57 | |

As shown in Table R1, when the 95% confidence intervals of the correlation between the Control and observation (e.g., r1) is not overlapped with the confidence intervals of the of the correlation between the $V_{thresh}12mn$ and observation (e.g., r2), it is considered the two correlations (r1 and r2) are significantly different. So the correlations for COD (Figs. 5 and 6) and fine dust concentration (Figs. 10) are significantly increased in the $V_{thresh}12mn$ simulation. In Fig. 9, although the correlation for the 16 RSMAS sites in the $V_{thresh}12mn$ actually decreases in comparison with that in the Control run (0.72 vs. 0.76), the differences with the observations are largely reduced (more white triangles, indicating more stations have the model to observation ratio between 0.5 and 2).

Seasonal cycles are shown in Figs. 11-13. Since the sample size is quite small, only 12 (months), the correlation can be less reliable; consequently the corresponding confidence intervals are quite large. We thus choose not to display correlations in plots but just list the correlations for each region/site in Table R2 here. As shown in Table R2, over most regions/sites correlations with the output from the $V_{thresh}12mn$ simulations increase in comparison with the correlations with the Control run.

Table R2 Correlations between the Control output and the observations (column 3) and between the $V_{thresh}12mn$ output with the observations (column 4) for Figs. 11-13. Correlation coefficients not significant at the 95% level are list in Italic.

| Figure # | Regions/sites | Correlation with Control | Correlation with $V_{thresh}12mn$ |
|---|---|---|---|
| Fig. 11 | Sahel | 0.90 | 0.86 |
| | Sahara | 0.81 | 0.94 |
| | Arabian Peninsula | 0.97 | 0.98 |
| | N. China | 0.44 | 0.58 |
| | India | 0.91 | 0.96 |
| | US | 0.91 | 0.90 |
| | S. Africa | *0.11* | *0.36* |
| | S. America | *0.40* | *0.54* |
| | Australia | 0.89 | 0.87 |
| Fig. 12 | Site 1 | 0.91 | 0.94 |
| | Site 2 | 0.88 | 0.65 |
| | Site 3 | 0.91 | 0.94 |
| | Site 4 | 0.99 | 0.98 |
| | Site 5 | 0.91 | 0.85 |
| | Site 6 | 0.90 | 0.93 |
| | Site 7 | 0.69 | 0.92 |
| | Site 8 | 0.82 | 0.95 |
| | Site 9 | 0.60 | 0.88 |
| | Site 10 | 0.67 | 0.83 |
| | Site 11 | 0.64 | 0.84 |
| | Site 12 | 0.73 | 0.80 |
| Fig. 13 | Banizoum | 0.72 | 0.90 |
| | Cinzana | 0.79 | 0.92 |
| | M'Bour | *0.14* | 0.92 |

Fig. 14 shows the case study, we choose not to apply correlation analysis, and the correlations for Fig. 15 are listed in Table R3. Over the Sahel, Arabian Peninsula, and

India the correlations are significantly higher than that between MODIS and the Control run.

Table R3 Correlations between model output and MODIS dust event frequency distribution as shown in Fig. 15. Correlation coefficients significant at the 95% confidence level are listed in bold. Whether the correlation between MODIS and the $V_{thresh}12mn$ simulation is significantly different from that between MODIS and the Control is indicated in the last column.

| Regions | Correlation with Control | Correlation with $V_{thresh}12mn$ | Significant? |
|---|---|---|---|
| Sahara | -0.17 | **0.62** | N |
| Sahel | -0.36 | **0.89** | Y |
| Arabian Peninsula | -0.28 | **0.96** | Y |
| N. China | -0.17 | 0.35 | N |
| US | -0.37 | -0.42 | N |
| India | 0.05 | **0.84** | Y |
| S. Africa | -0.30 | -0.33 | N |
| S. America | -0.13 | -0.15 | N |
| Australia | 0.35 | 0.42 | N |

*Other comments:*
*- Line 2: I'd suggest saying "many" instead of "most", as I believe most models at least account for the effect of soil moisture on the threshold wind speed.*
    Done.

*- Do you have a sense of how sensitive your results are to the particular reanalysis product used?*
    We tested the sensitivities of our method to surface winds in different reanalysis products in Table 3. Discussion is added in section 2.3.

*- Line 304: it seems hard to imagine that snow cover of 0.2% would prevent or substantially reduce the occurrence of wind erosion. Please provide support for this assumption.*
    We added in line 342-343: "since snow cover percentage is round-up to integer in MODIS product, this criterion actually requires no snow cover". We also test the sensitivities of the results to the criteria of snow cover and found it only slightly affect the magnitude of the threshold wind speed in a few regions (Table 2), such as northern China, U.S., and South America, by up to 0.3 m s$^{-1}$ if changing from no snow cover to 10%.

*- Line 311-2: "soil moisture ranging from 1.01 to 11.2 kg kg-3"; the units here are incorrect, and I think the number is much too high if the intended unit was kg of water per kg of soil.*
    Thanks for pointing this out. The numbers are from Table 1 in (Fećan et al. 1999), and the unit is %.

*- Line 317: I don't think it makes sense to only pick out the daily maximum surface wind speed when you have wind speeds at 6-hours resolution. You could either argue that the DOD is a product of emission that occurs over a longer time period and thus use winds at all time steps, or you could argue that you are using DOD as a proxy for emissions in the moment and thus use the wind speed closest to your DOD observation (presumably noon since overpasses are at 10:30 am and 1:30 pm). But using the daily maximum does not make sense to me.*

We use daily maximum wind speed largely because wind erosion occurs when the wind speed is relatively high, so we want to focus on the maximum of 6-hourly wind speed. Ginoux and Deroubaix (2017) also used daily maximum surface wind speed from the EAR-Interim to retrieve threshold of wind erosion over northern China. We added line 361-365 to better clarify this point: "Following Ginoux and Deroubaix (2017), we use maximum daily wind speed instead of daily mean wind speed, largely because dust emission only occur when wind speed is strong enough, and the emission magnitude is roughly proportional to the third power of surface wind speed in empirical estimations."

*- The authors use a threshold DOD of 0.2 over the major source regions of North Africa, the Middle East, etc, which is consistent with previous work in Ginoux et al. (2012). But they use a threshold DOD of only 0.02 in lesser source regions such the US, South America, etc. This is a very large difference of a factor of 10, and seems rather arbitrary. Could the authors either provide an analysis of the sensitivity of their results to this choice or use the actual frequency distribution of DOD in the different source regions to inform these thresholds?*

As shown in Fig. 15 and also mentioned in lines 376-378 and 748-751, the regional mean DOD frequency distribution in less dusty regions, such as the U.S., South America, South Africa, Australia, peaks at a much lower value. We chose to use a DOD threshold ten times smaller for these less dusty regions also because the magnitude of dust emission in these regions are at least ten times smaller than major dust source regions such as North Africa and the Middle East. In Table 2 (also see discussion in section 2.3), we tested the sensitivity of using $DOD_{thresh}$ of 0.5 and 0.05. In the U.S., South Africa, South America, and Australia, changing $DOD_{thresh}$ from 0.02 to 0.05 will increase annual mean threshold of wind erosion by about 1.27, 1.05, 1.74 and 1.30 m s$^{-1}$, respectively.

*- Section 3.2.3: How are you obtaining AERONET data as a gridded product since data density is so sparse in most dust source regions?*

In this section, we shows regional averaged DOD from model output along with MODIS DOD and grided AERONET COD (interpolating from station data to a 0.5° by 0.5° grid) in Fig. 11, while in Figs. 12-13, only AERONET station data are shown. AERONET station data are quite sparse in some regions, e.g., the Sahel, thus the interpolated COD has a large difference with MODIS. So when discussing Fig. 11 we mentioned in lines 665-668: "Since the gridded COD may have large uncertainties over regions with only a few stations, such as the Sahel, Sahara, northern China, and South Africa, MODIS DOD is used as the main reference in the comparison."

*- It's not clear to me whether the control run accounts for the effects of soil moisture on the threshold wind speed or whether it truly uses a constant threshold wind speed, regardless even of soil moisture content. Could you clarify?*

The default setting in the AM4.0/LM4.0, or the Control run, does not include soil moisture in dust emission. The dust emission scheme follows Eq. 4. So a constant threshold of wind erosion is used.

*Editorial comments:*
*- Line 57: Since wind speeds are a function of height, please note what these wind speeds refer to.*

We added "for surface 10 m wind" in line 57.

*- Since the methodology is quite involved and lengthy, I recommend you provide an overview of your methodology in a paragraph at the beginning of section 2 to make the paper easier to read.*

Thanks for your suggestion. We added lines 124-129 to better introduce this section.

*- 182-184: Please provide more info or a citation to a peer-reviewed paper here for the reader to understand how LAI is calculated.*

We added reference in text (Yan et al. 2016a, b).

*- Line 254-5: This is a common assumption in using the dust concentration data, so you could support this by citing precedent in previous studies.*

We modified lines 289-292 to better clarify this point. We cited Ginoux et al. (2001) and Huneeus et al. (2011) who also used the data to validate model output in different periods in lines 281-283.

*- Section 3.3: I think this section would be placed more logically before the case study.*

We'd like to keep the original order because section 3.2.4 is a case study about the DOD simulation in one region at a particular time (a few days), while section 3.4 examined global frequency distribution of DOD in the model, an aspect largely ignored by previous studies. So we'd like to keep it in a separate section. Also, in section 3.3, DOD frequency distributions in MODIS, the Control and $V_{thresh}12mn$ simulations are discussed and summarized for individual regions.

*- Figure 8: since the data here span 3 orders of magnitude, providing statistics in linear space is not very meaningful as it weighed heavily toward the large concentration data. Please provide statistics in logarithmic space.*

We updated Fig. 9 (previously Fig. 8) to change the statistics in logarithmic space.

*- Fig. 14: What is the bin spacing on the horizontal axis? The reader needs that to interpret the percentage given on the vertical axis.*

We added the information in figure caption. The bin spacing for dusty regions is 0.05 while for less dusty regions is 0.01.

[revised manuscript text omitted]

---

## Author Response (AR2)

Department of Geography and Atmospheric Science
University of Kansas
Jayhawk Boulevard
Lawrence, KS 66045
Email: bpu@ku.edu

Dr. Yves Balkanski
Institut Pascal et
IPSL/LSCE (Laboratoire des Sciences du Climat et de l'Environnement)
CEA-CNRS-UVSQ-UPSaclay UMR 8212
L'Orme des Merisiers - Bat 714, pce 1012
91191 Gif sur Yvette Cedex, FRANCE

November 21[st], 2019

Dear Editor Balkanski,

Your helpful comments are sincerely appreciated. We have submitted a revised paper entitled "Retrieving the global distribution of threshold of wind erosion from satellite data and implementing it into the GFDL AM4.0/LM4.0 model" by B. Pu, P. Ginoux and co-authors to address these comments. We also made a few minor edits in the text.

We gratefully appreciate your time and consideration!

Sincerely,

Bing Pu

We sincerely appreciate the helpful comments from the Co-Editor. We reply to your comment (in Italic) below.

*Comments to the Author:*
*The authors have adequately addressed the main concerns from the 2 reviewers by doing the additional sensitivity tests on soil moisture, LAI, snow coverage and threshold dust optical depth summarized in Table 2.*
*I would like the authors to add a sentence in the caption of Table 2 to explain the 2 different thresholds for DOD 0.2 (0.02) and 0.5 (0.05) depending on regions. This will help the reader understand this notation.*

We added "Here $DOD_{thresh}$=0.2 or 0.5 is applied to dusty regions, i.e., the Sahel, Sahara, Arabian Peninsula, northern China, and India, while $DOD_{thresh}$=0.02 or 0.05 is applied to less dusty regions, i.e., the U.S., South Africa, South America, and Australia." To the caption of Table 2.

*Now that Figure 9 (ex Fig. 8) is in log-scale, the correlation factors have changed, please check the value for these correlation factors.*

Thanks for noticing this. We checked the correlation coefficients and other statistics in Fig. 9, and they are based on log-scale data. The correlation coefficient in the Control run (top left plot) looks the same as the previous version due to the precision level applied here. They are actually slightly different: the original correlation coefficient is 0.762638, and the correlation for the log-scale data is 0.756081.

[revised manuscript text omitted]